# Upper airway gene expression shows a more robust adaptive immune response to SARS-CoV-2 in children

Eran Mick [1,2,3,8], Alexandra Tsitsiklis [1,8], Natasha Spottiswoode[1], Saharai Caldera[1,3], Paula Hayakawa Serpa[1,3], Angela M. Detweiler[3], Norma Neff [3], Angela Oliveira Pisco [3], Lucy M. Li[3], Hanna Retallack[4], Kalani Ratnasiri [3], Kayla M. Williamson[5], Victoria Soesanto[5], Eric A. F. Simões[6], Christiana Smith[6], Lisa Abuogi[6], Amy Kistler[3], Brandie D. Wagner[5,6], Joseph L. DeRisi[3,4], Lilliam Ambroggio[6], Peter M. Mourani[6,7,9] & Charles R. Langelier [1,3,9✉]

Unlike other respiratory viruses, SARS-CoV-2 disproportionately causes severe disease in older adults whereas disease burden in children is lower. To investigate whether differences in the upper airway immune response may contribute to this disparity, we compare nasopharyngeal gene expression in 83 children (<19-years-old; 38 with SARS-CoV-2, 11 with other respiratory viruses, 34 with no virus) and 154 older adults (>40-years-old; 45 with SARS-CoV-2, 28 with other respiratory viruses, 81 with no virus). Expression of interferon-stimulated genes is robustly activated in both children and adults with SARS-CoV-2 infection compared to the respective non-viral groups, with only subtle distinctions. Children, however, demonstrate markedly greater upregulation of pathways related to B cell and T cell activation and proinflammatory cytokine signaling, including response to TNF and production of IFNγ, IL-2 and IL-4. Cell type deconvolution confirms greater recruitment of B cells, and to a lesser degree macrophages, to the upper airway of children. Only children exhibit a decrease in proportions of ciliated cells, among the primary targets of SARS-CoV-2, upon infection. These findings demonstrate that children elicit a more robust innate and especially adaptive immune response to SARS-CoV-2 in the upper airway that likely contributes to their protection from severe disease in the lower airway.

[1] Division of Infectious Diseases, University of California, San Francisco, CA, USA. [2] Division of Pulmonary and Critical Care Medicine, University of California, San Francisco, CA, USA. [3] Chan Zuckerberg Biohub, San Francisco, CA, USA. [4] Department of Biochemistry and Biophysics, University of California, San Francisco, CA, USA. [5] Department of Biostatistics and Informatics, Colorado School of Public Health, University of Colorado, Aurora, CO, USA. [6] Department of Pediatrics, University of Colorado and Children's Hospital Colorado, Aurora, CO, USA. [7] Present address: Arkansas Children's Research Institute, Arkansas Children's Hospital, Little Rock, AR, USA. [8] These authors contributed equally: Eran Mick, Alexandra Tsitsiklis. [9] These authors jointly supervised this work: Peter M. Mourani, Charles R. Langelier. ✉email: chaz.langelier@ucsf.edu

One of the defining features of the COVID-19 pandemic has been the striking relationship between disease severity and age[1–3]. While infection with other respiratory viruses, such as influenza or respiratory syncytial virus, causes significant morbidity and mortality in both young children and older adults[4–9], severe COVID-19 occurs disproportionately in older adults[1–3,10–14]. A comprehensive modeling study, conducted prior to the advent of vaccines, estimated that the infection fatality rate was lowest for children ages 5–9 (~0.001%) and that even adults in their 40 s were already at 100-fold greater risk of death from COVID-19[1]. The age-dependent effect on disease severity and mortality has been shown even when accounting for age-associated comorbidities[15].

A few studies have examined differences in systemic immunological profiles of children and adults with COVID-19. Among patients in the early/mild phase of infection, children and adults displayed a comparable magnitude of systemic innate immune responses, including interferon-stimulated and pro-inflammatory gene expression[16]. These responses tended to resolve more quickly in children, however, while adults maintained a prolonged inflammatory and cytotoxic response in the circulation[16–18]. Among hospitalized patients, adults displayed greater breadth and neutralizing activity of SARS-CoV-2 specific antibodies as well as stronger CD4+ T cell responses to spike protein[19,20], suggesting that poorer outcomes in adults are not due to failure to engage a robust systemic adaptive response and may even be driven by it.

Recent work has also begun to shed light on age-related differences in the immune response at the site of initial infection, the upper airway. It has been proposed that the upper airway of children is primed for viral sensing, exhibits a pre-activated anti-viral state, and/or engages a more robust innate immune response upon SARS-CoV-2 infection[17,21,22]. However, numerous studies have found little to no evidence of a systematic difference between infected children and adults in the distribution of SARS-CoV-2 viral load in the nasopharynx or in the kinetics of viral clearance[16,23–26], and a few studies have even shown infants exhibit the highest viral load[27,28]. This suggests children are not significantly better able to achieve early control of viral replication in the upper airway. Nevertheless, differences in the upper airway microenvironment and immune response could contribute to protection from severe disease in children in additional ways, for example, by limiting migration of the virus into the lower airway.

Several studies have placed particular focus on potential differences in interferon-stimulated gene (ISG) expression in the upper airway of children and adults, given its well-established importance as a front-line of anti-viral innate immunity[17,21,22,29]. However, these studies reported some contradictory results, and none directly controlled for SARS-CoV-2 viral load on a gene-by-gene basis, highlighting the need for further investigation.

Here, we assess age-related differences in upper airway gene expression in response to SARS-CoV-2 infection by comparing previously published RNA-sequencing data of nasopharyngeal (NP) swabs from an adult cohort[30] with new sequencing data from a pediatric cohort. Our results suggest that differences in the overall magnitude of ISG expression in the upper airway of children and adults with COVID-19 are subtle and appear unlikely to explain their distinct clinical outcomes. However, we also find evidence of more robust innate and especially adaptive immune responses in the upper airway of children, as well as increased clearance of ciliated cells, which may contribute to their protection from severe disease.

## Results

To compare the upper airway gene expression response to SARS-CoV-2 infection in children and adults, we utilized a previously published dataset of NP swab RNA-sequencing from an adult cohort[30] alongside newly sequenced swabs from a pediatric cohort. All samples were obtained in the course of clinical testing for SARS-CoV-2 infection at the University of California San Francisco or Children's Hospital Colorado between March and September 2020, prior to the availability of COVID-19 vaccines. We included patients up to 19 years of age in the pediatric cohort and restricted the adult cohort to those at least 40 years of age to impose clearer separation.

We divided each age cohort into three viral status groups: 1) patients with PCR-confirmed SARS-CoV-2 infection ("SARS-CoV-2" group), 2) patients negative for SARS-CoV-2 by PCR with no other pathogenic respiratory virus detected by metagenomic RNA sequencing ("No Virus" group), and 3) patients negative for SARS-CoV-2 who had another respiratory virus detected by sequencing ("Other Virus" group). Finally, we limited the samples in the SARS-CoV-2 group to those with at least 10 viral reads-per-million (rpM), comparable to PCR $C_t$ values below 30[30]. Viral load above this threshold is characteristic of acute infection, from just before symptom onset up to ~6 days later[31], and has been associated with recovery of actively replicating virus[32–34].

The final dataset included 83 children (38 SARS-CoV-2, 34 No Virus, 11 Other Virus; median age 4 years, IQR 2-12) and 154 adults (45 SARS-CoV-2, 81 No Virus, 28 Other Virus; median age 62 years, IQR 47-71) (Fig. 1a, b; Table 1; Supplementary Data 1). Most of the patients in the SARS-CoV-2 group in both age cohorts were tested as outpatients, indicative of an early/mild stage of disease (Table 1). Samples in the SARS-CoV-2 group in both age cohorts spanned several orders of magnitude of viral load, and while viral load trended higher in the children, this did not reach statistical significance (Fig. 1c). Patients in the No Virus group in both age cohorts were more likely to be hospitalized, with a higher proportion in the pediatric cohort (Table 1; Supplementary Data 1). Rhinovirus was the most prevalent among the other respiratory viruses in both age cohorts (Fig. 1d).

We began by performing differential expression (DE) analyses between the SARS-CoV-2 and No Virus groups within each age cohort separately. This approach minimizes confounding by age differences unrelated to SARS-CoV-2 infection and by any potential batch effects, though it could be influenced by differences between each cohort's No Virus group. The analyses yielded 1,961 and 1,216 differentially expressed genes at a $p$-value < 0.1 (based on a moderated $t$-statistic and Benjamini-Hochberg adjusted) for the pediatric and adult cohorts, respectively (Supplementary Data 2). As expected, interferon-stimulated genes (ISGs) were prominent among the genes highly significant in both age cohorts (Supplementary Fig. 1a). Overall, children exhibited a considerably larger number of unique DE genes, including immune-related genes, despite a smaller sample size that would have been expected to provide less statistical power (Supplementary Fig. 1a).

We next performed gene set enrichment analyses[35] (GSEA) using Gene Ontology (GO) biological process annotations[36] on the DE results from each cohort and compared the enriched pathways. As expected, a range of immune-related pathways were upregulated in both adults and children with SARS-CoV-2 infection compared to those with no virus (Fig. 2a; Supplementary Fig. 1b; Supplementary Data 3). Pathways related to the interferon response appeared as a whole to be strongly induced in both children and adults. Children, however, demonstrated stronger upregulation of the pathways for B cell activation, T cell activation, response to TNF, macrophage activation and phagocytosis. Children also exhibited stronger activation of several cytokine production pathways typically associated with T cell activation, such as IL-2, IL-4, and IFNγ production (Supplementary Fig. 2a–c). A few pathways showed expression changes

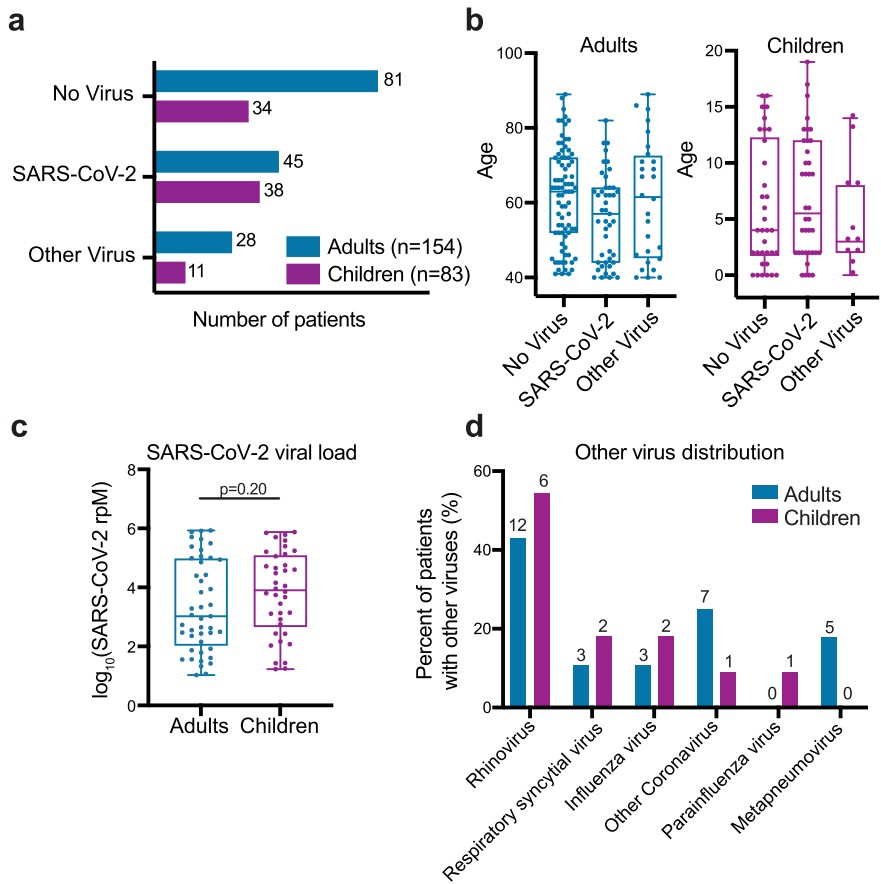

**Fig. 1 Patient numbers, age distribution, SARS-CoV-2 viral load and other viruses present in the adult and pediatric cohorts. a** Number of patients in the SARS-CoV-2, No Virus and Other Virus groups in the adult and pediatric cohorts. Color indicates the age cohort. **b** Age distribution across the three viral status groups in the adult and pediatric cohorts. Horizontal lines denote the median, box boundaries represent the interquartile range, and whiskers extend to minimum and maximum. Adults-No Virus $n = 81$, Adults-SARS-CoV-2 $n = 45$, Adults-Other Virus $n = 28$, Children-No Virus $n = 34$, Children-SARS-CoV-2 $n = 38$, Children-Other Virus $n = 11$. **c** Distribution of SARS-CoV-2 viral load, measured in reads-per-million (rpM), in adult ($n = 45$) and pediatric ($n = 38$) patients. Horizontal lines denote the median, box boundaries represent the interquartile range, and whiskers extend to minimum and maximum. *P*-value derives from a two-sided Mann-Whitney test. **d** Distribution of viruses in the Other Virus groups in the adult and pediatric cohorts. Absolute numbers are provided above each bar, the y-axis indicates percentage out of each cohort's Other Virus group. One child was infected with both influenza and rhinovirus and two adults were infected with both respiratory syncytial virus and rhinovirus.

in opposite directions compared to the respective No Virus group, such as neutrophil mediated immunity and mast cell mediated immunity. Both children and adults infected with SARS-CoV-2 exhibited downregulation of olfactory receptor gene expression ('sensory perception of chemical stimulus'), consistent with the loss of sense of smell that has been clinically observed across the age spectrum[37,38].

We complemented the analyses comparing SARS-CoV-2 and No Virus patients in each cohort separately by directly comparing gene expression between SARS-CoV-2 infected children and adults, controlling for viral load. We identified 5,352 differentially expressed genes at a *p*-value < 0.1 (based on a moderated *t*-statistic and Benjamini-Hochberg adjusted; Supplementary Data 4). Age differences unrelated to viral status likely contributed to the larger number of DE genes in the direct comparison. Nevertheless, GSEA of the DE results yielded overall similar patterns to those described above with regard to immune pathways (Fig. 2b; Supplementary Data 5). B cell related pathways (B cell activation, humoral immune response), T cell related pathways (T cell activation, IL-2, IL-4 and IFNγ production) and chemokine/cytokine signaling were more highly expressed in children with SARS-CoV-2 infection.

While some immune pathways did not reach statistical significance in the direct comparison between children and adults with SARS-CoV-2 infection, they typically trended in the same direction observed in the comparison to the No Virus groups (Fig. 2a, b). On the other hand, the stark disparity in neutrophil activation observed in the comparison to the No Virus groups was only weakly supported in the direct comparison, likely reflecting differences among the No Virus patients themselves. The direct comparison clearly revealed lower expression of cilia-associated genes in children with SARS-CoV-2 infection and suggested a trend toward lower expression of interferon-stimulated genes (Fig. 2b; Supplementary Fig. 2d), though the pathway just missed the statistical significance cutoff (*p*-value = 0.06, based on an adaptive multilevel splitting Monte Carlo approach and Benjamini-Hochberg adjusted). As expected, many developmental processes unrelated to infection also differed in the direct comparison between children and adults (Supplementary Data 5).

Importantly, we observed similar DE and GSEA results in a secondary analysis restricted only to outpatient children ($n = 30$) and adults ($n = 24$) with SARS-CoV-2 infection, suggesting that differences in the proportion of hospitalized patients and

**Table 1 Adult and Pediatric cohort characteristics.**

| | Adult Cohort | | | | | Pediatric Cohort | | | | | Adult vs Pediatric | |
|---|---|---|---|---|---|---|---|---|---|---|---|---|
| | Cohort Overall | SARS-CoV-2 | Other Virus | No Virus | p-value | Cohort Overall | SARS-CoV-2 | Other Virus | No Virus | p-value | Overall p-value | SARS-CoV-2 p-value |
| Total Enrolled (n) | 154 | 45 | 28 | 81 | | 83 | 38 | 11 | 34 | | | |
| Age, years (median, range, (QR) | 62 (40–89, 47–71) | 57 (40–82, 44–64) | 61.5 (40–89, 45–73) | 62 (41–89, 52–72) | 0.07 | 4 (<1–19, 2–12) | 5 (<1–19, 2–11) | 3 (<1–14, 2–11) | 4 (<1–16, 2–12) | 0.97 | | 0.27 |
| Female gender | 78 (51%) | 25 (56%) | 10 (36%) | 43 (53%) | 0.21 | 42 (51%) | 16 (42%) | 7 (64%) | 19 (56%) | 0.38 | 0.93 | 0.27 |
| Clinical Encounter Type | | | | | 0.003 | | | | | <0.001 | 0.59 | 0.23 |
| Inpatient | 42 (27%) | 4 (9%) | 9 (32%) | 29 (36%) | | 31 (37%) | 1 (3%) | 2 (18%) | 28 (82%) | | | |
| Intensive Care Unit | 16 (10%) | 2 (4%) | 5 (18%) | 9 (11%) | | 0 (0%) | 0 (0%) | 0 (0%) | 0 (0%) | | | |
| Emergency Department | 24 (16%) | 3 (7%) | 6 (21%) | 15 (19%) | | 19 (23%) | 6 (16%) | 8 (73%) | 5 (15%) | | | |
| Outpatient | 55 (36%) | 24 (53%) | 8 (29%) | 23 (28%) | | 31 (37%) | 30 (79%) | 0 (0%) | 1 (3%) | | | |
| Unknown | 17 (11%) | 12 (27%) | 0 (0%) | 5 (6%) | | 2 (2%) | 1 (3%) | 1 (9%) | 0 (0%) | | | |
| Race | | | | | <0.001 | | | | | 0.31 | 0.013 | 0.1 |
| White or Caucasian | 67 (44%) | 6 (13%) | 19 (68%) | 42 (52%) | | 34 (41%) | 13 (34%) | 4 (36%) | 17 (50%) | | | |
| Asian | 25 (16%) | 6 (13%) | 5 (18%) | 14 (17%) | | 6 (7%) | 1 (3%) | 1 (9%) | 4 (12%) | | | |
| Black or African American | 14 (9%) | 1 (2%) | 1 (4%) | 12 (15%) | | 4 (5%) | 2 (5%) | 2 (18%) | 0 (0%) | | | |
| Native Hawaiian or Pacific Islander | 1 (1%) | 1 (2%) | 0 (0%) | 0 (0%) | | 2 (2%) | 0 (0%) | 1 (9%) | 1 (3%) | | | |
| American Indian or Alaska Native | 0 (0%) | 0 (0%) | 0 (0%) | 0 (0%) | | 1 (1%) | 0 (0%) | 0 (0%) | 1 (3%) | | | |
| Other | 27 (18%) | 17 (38%) | 3 (11%) | 7 (9%) | | 29 (35%) | 17 (45%) | 2 (18%) | 10 (29%) | | | |
| Unknown | 20 (13%) | 14 (31%) | 0 (0%) | 6 (7%) | | 7 (8%) | 5 (13%) | 1 (9%) | 1 (3%) | | | |
| Ethnicity | | | | | <0.001 | | | | | <0.001 | <0.001 | 0.06 |
| Not Hispanic or Latino | 109 (71%) | 16 (36%) | 26 (93%) | 67 (83%) | | 43 (52%) | 10 (26%) | 7 (64%) | 26 (76%) | | | |
| Hispanic or Latino | 23 (15%) | 14 (31%) | 1 (4%) | 8 (10%) | | 34 (41%) | 23 (61%) | 3 (27%) | 8 (24%) | | | |
| Unknown | 22 (14%) | 15 (33%) | 1 (4%) | 6 (7%) | | 6 (7%) | 5 (7%) | 1 (9%) | 0 (0%) | | | |

Values are n (%) unless otherwise indicated.
Race categories with <1 patient in ≥ 1 group were excluded from analyses.
Unknown values were excluded from analyses.
Age was analyzed as one-way ANOVA and all categorial variables were analyzed using chi-square.

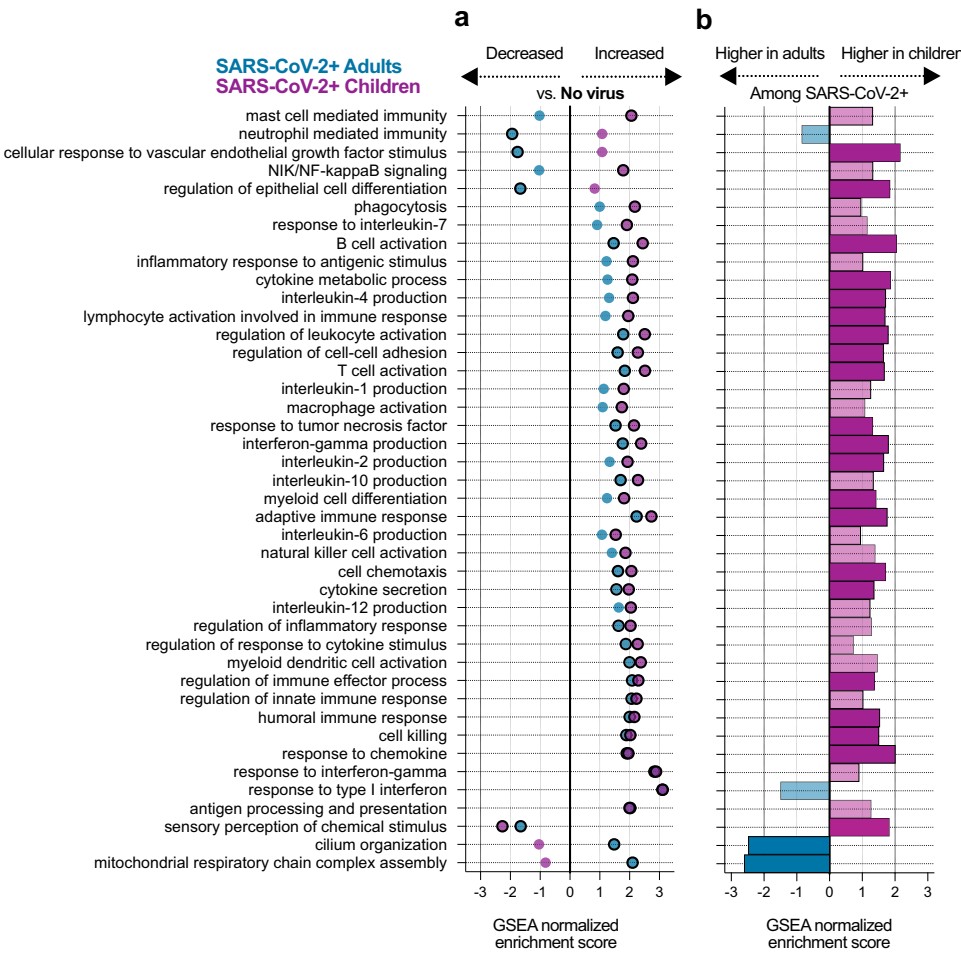

**Fig. 2 Pathways activated in children and adults upon SARS-CoV-2 infection. a** Normalized enrichment scores of selected GO biological process terms that reached statistical significance (adjusted *p*-value < 0.05) in the gene set enrichment analysis (GSEA) using differentially expressed (DE) genes between the SARS-CoV-2 and No Virus groups in either the adult or pediatric cohort. Color indicates the age cohort. Pathway *p*-values were calculated using an adaptive, multilevel splitting Monte Carlo approach and Benjamini–Hochberg adjusted, and statistical significance is denoted by a black outline around the circle. Complete results are provided in Supplementary Data 3. **b** Normalized enrichment scores for the same GO terms as in (**a**) in the GSEA using DE genes between children and adults with SARS-CoV-2 infection. Dark color bars represent pathways that reached statistical significance (adjusted *p*-value < 0.05). Complete results are provided in Supplementary Data 5.

outpatients in each cohort did not bias the direct comparison (Supplementary Fig. 3).

Many of the pathways identified in the GSEA results as differentially expressed between children and adults with SARS-CoV-2 infection were tightly related to specific cell types. We therefore applied in silico estimation of cell type proportions[39] based on marker genes derived from an airway single-cell study[40] as an additional approach to contextualize our findings (Fig. 3; Supplementary Fig. 4; Supplementary Data 6). Consistent with the GSEA results, we found that SARS-CoV-2 infection triggered significantly greater recruitment of B cells to the upper airway in children compared to adults, which was also evident in the comparison between children and adults with other respiratory viruses (Fig. 3a). In contrast, differences in estimated T cell proportions were much subtler (Supplementary Fig. 4a), suggesting the GSEA results may reflect distinctions in T cell identity and regulation and not only cell number.

In the previous analysis of the adult cohort, infection with SARS-CoV-2 was associated with blunted recruitment of macrophages and neutrophils to the upper airway as compared to other respiratory viral infections[30]. The pediatric Other Virus group was too small to definitively conclude whether this

observation recapitulates in children, especially given the mix of different viruses represented. Nevertheless, it is notable that a substantial fraction of the samples in the pediatric Other Virus group indeed exhibited markedly higher macrophage, neutrophil and dendritic cell proportions than most SARS-CoV-2 samples (Fig. 3b–d). Macrophage proportions did trend higher in children with SARS-CoV-2 infection compared with adults (Fig. 3b).

Intriguingly, while proportions of ciliated cells did not differ between children and adults in the No Virus groups, children with SARS-CoV-2 infection exhibited a marked decrease in ciliated cell proportions that was absent in the adults (Fig. 3e), consistent with the GSEA findings. This was accompanied by greater proportions of basal cells in children compared to adults with SARS-CoV-2 infection (Fig. 3f), possibly reflecting compensatory regeneration of the airway epithelium.

Finally, we wished to examine the effect of SARS-CoV-2 viral load on gene expression in pathways of interest within each age cohort. Expression of interferon-stimulated genes (ISGs) frequently correlates with viral load, as was previously observed in the adult cohort[30]. We performed robust regression to relate the expression of *n* = 100 ISGs to viral load in children or adults with SARS-CoV-2 infection, and compared the resulting slopes and

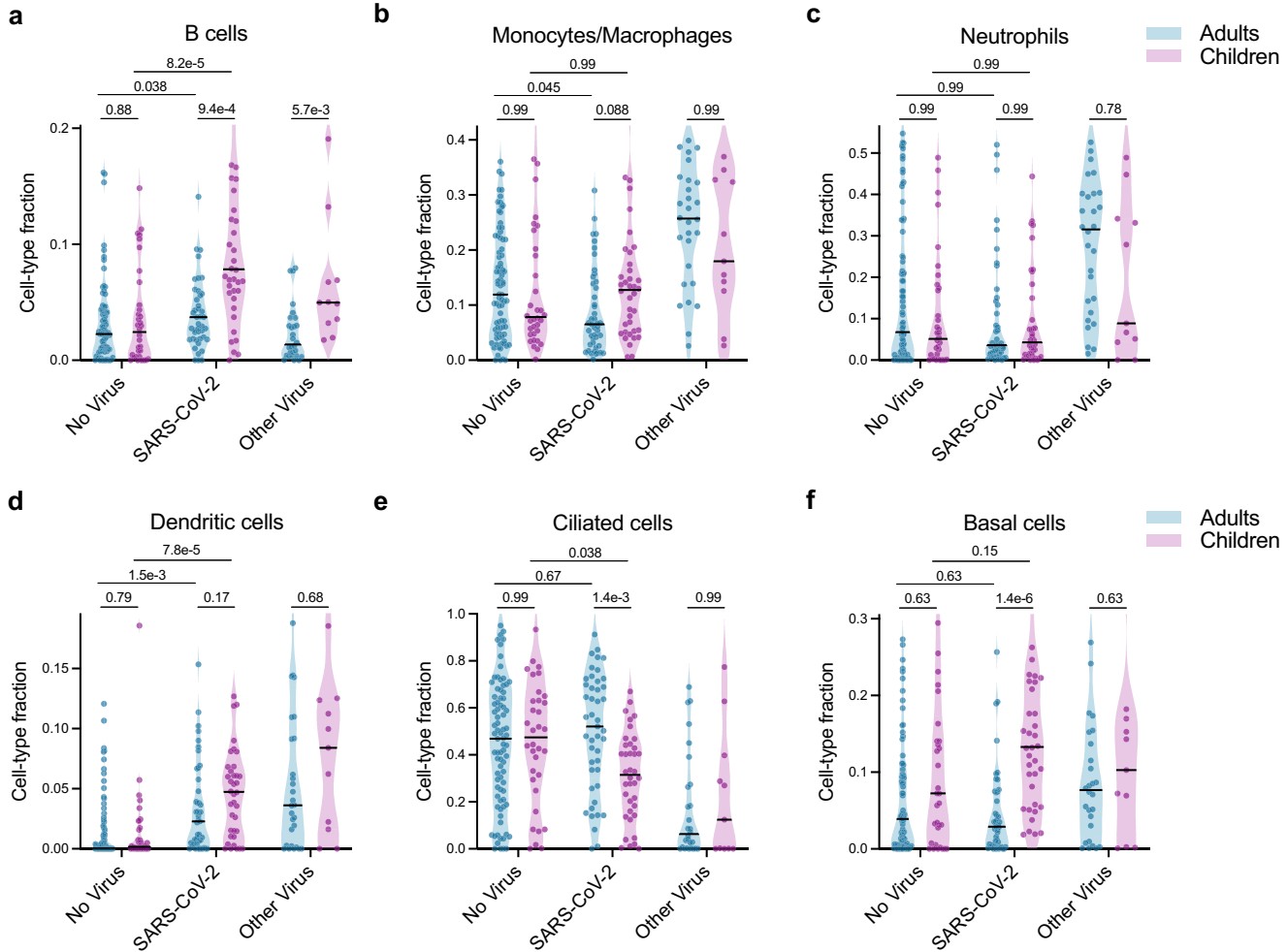

**Fig. 3 Cell-type proportion differences between children and adults. a–f** In silico estimation of cell-type proportions in the bulk RNA-sequencing using single-cell signatures. Color indicates the age cohort. Black lines denote the median. The y-axis in each panel was trimmed at the maximum value among all groups of 1.5*IQR above the third quartile, where IQR is the interquartile range. For each cell type, we formally compared each viral status group between the two age cohorts as well as the No Virus and SARS-CoV-2 groups within each age cohort. Pairwise comparisons were performed with a two-sided Mann-Whitney test followed by Holm's correction for multiple testing. Adults-No Virus $n = 81$, Adults-SARS-CoV-2 $n = 45$, Adults-Other Virus n = 28, Children-No Virus $n = 34$, Children-SARS-CoV-2 $n = 38$, Children-Other Virus $n = 11$. Complete results are provided in Supplementary Data 6.

coefficients of determination (Fig. 4a, b). The ISGs that most strongly correlated with viral load in adults, such as *CXCL11* and *OASL*, exhibited overall similar patterns in children with slightly greater slopes (Fig. 4a–c). However, a subset of ISGs was considerably better correlated with viral load in children, most strikingly exemplified by genes such as *IFI6* and *IFI27* (Fig. 4d). While even adults with low viral load displayed elevated expression of these genes, the response in children was more gradual and only caught up to the adults at higher viral loads. ISGs that shifted from an almost stepwise response to the virus in adults to a more proportional one in children were among the leading-edge genes that contributed to the apparent trend toward lower interferon-response pathway expression in children in the GSEA results (Fig. 2b; Supplementary Fig. 2d; Supplementary Data 5). These findings suggest relatively subtle differences in ISG-specific regulation and/or cellular origins between children and adults that defy simple generalization. Such regulatory differences might also be reflected in the differential expression of certain interferon-regulatory factors (e.g., *IRF8*; Supplementary Fig. 2c).

In stark contrast to ISGs, the expression of B cell marker genes, such as *CD22* and *CD79A*, was entirely uncorrelated with viral load in children (Fig. 4d). These genes exhibited significant

heterogeneity between patients, likely reflecting the timing of activation of the B cell response, but the fraction of children who were engaging the response at the time of sampling was substantially greater.

## Discussion

We compared upper airway gene expression in children and adults to identify commonalities and distinctions in the response to SARS-CoV-2 at the site of initial infection, which may ultimately contribute to their disparate clinical outcomes.

Our analysis supports the conclusion that, when controlling for viral load, children and adults with SARS-CoV-2 infection both engage a pronounced interferon-stimulated gene (ISG) response in the upper airway that, as a whole, is of comparable magnitude. Our data further demonstrate that children exhibit a more gradual and proportional 'dose response' to viral load for a subset of prominent ISGs. These results are broadly in line with the findings of Koch et al., who also performed bulk RNA-sequencing on upper airway samples from children and adults and assessed a composite measure of ISG expression in patients with the highest viral load[29]. Yoshida et al. who performed a single-cell RNA-sequencing study, also observed only subtle distinctions, with

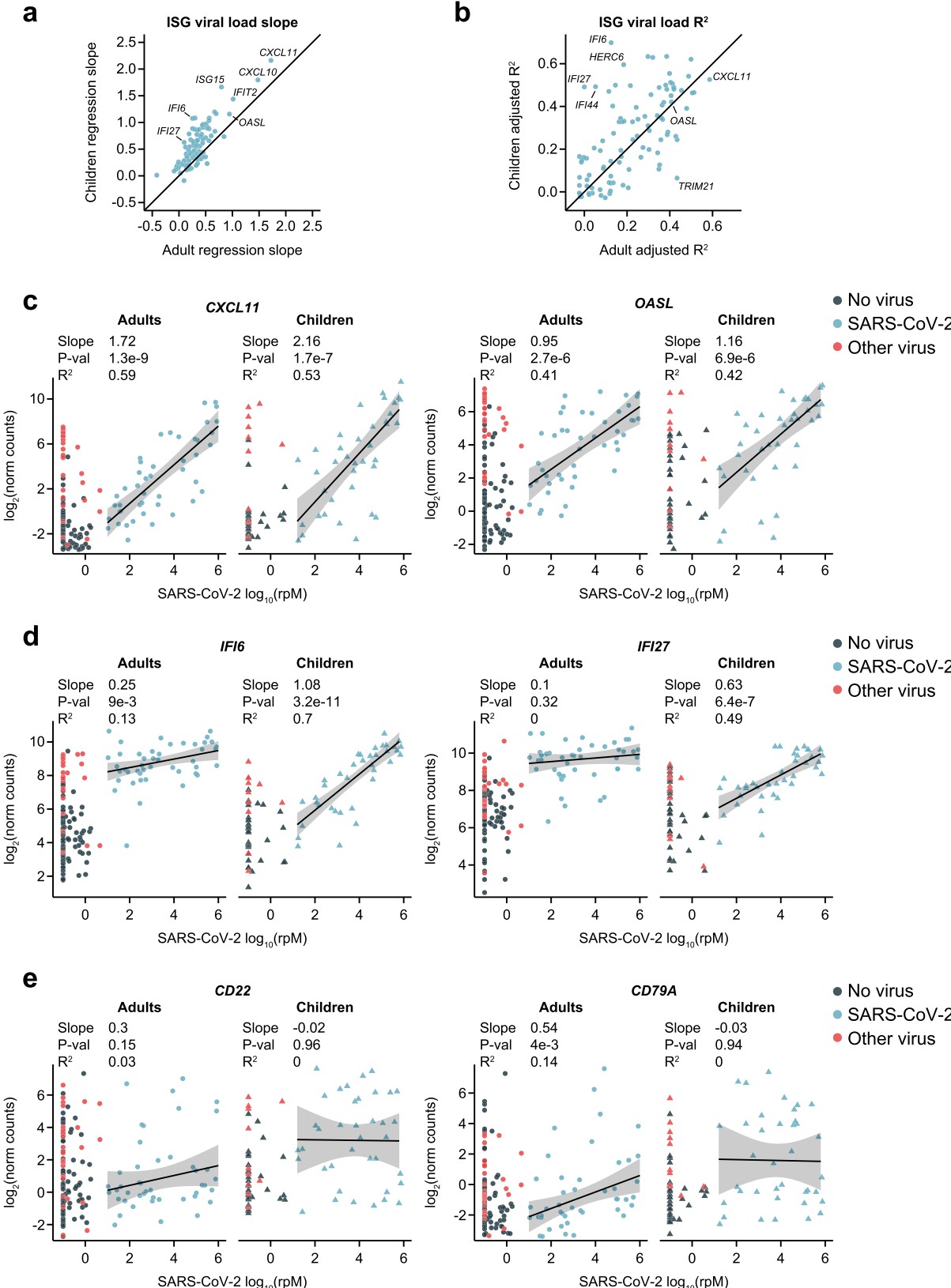

slightly stronger upregulation of composite ISG expression in epithelial cells of infected adults but slightly stronger upregulation in immune cells of infected children[17]. In contrast, the single-cell analysis by Loske et al. showed somewhat elevated ISG expression in children, though neither single-cell study directly controlled for viral load[21].

Importantly, both single-cell studies suggested a pre-activated anti-viral state in healthy children, characterized by elevated expression of upstream viral pattern recognition receptors and/or ISGs themselves[17,21]. Our study was not well suited to examine this question since the patients in the No Virus groups were generally not healthy controls. It may indeed be the case that a

**Fig. 4 Relationship of SARS-CoV-2 viral load to ISG and B cell marker gene expression in children and adults. a** Scatter plot of the slopes from robust regression of the expression of 100 interferon-stimulated genes (ISGs) against viral load in adults ($n = 45$; x-axis) and children ($n = 38$; y-axis) with SARS-CoV-2 infection. **b** Scatter plot of the adjusted coefficients of determination ($R^2$) from robust regression of the expression of 100 ISGs against viral load in adults ($n = 45$; x-axis) and children ($n = 38$; y-axis). **c** Scatter plots of normalized gene counts ($\log_2$ scale, y-axis) as a function of SARS-CoV-2 viral load ($\log_{10}$(rpM), x-axis) in each age cohort for canonical type I interferon response genes showing high correlation to viral load in adults and children. The viral status group is indicated by the dot color. Robust regression was performed on SARS-CoV-2 patients to characterize the relationship to viral load in each age cohort ($n = 45$ adults, $n = 38$ children). Shaded bands represent 95% confidence intervals around the mean predicted value. Numerical results listed for each gene refer to, from top to bottom: the regression slope, the nominal $p$-value for the difference of the slope from 0 (derived from a $t$-statistic), and the adjusted robust coefficient of determination ($R^2$). **d** Plots as in **c**, for ISGs showing a more gradual response to viral load in children. **e** Plots as in **c**, for B cell marker genes.

pre-activated anti-viral state decreases the chance that a productive infection is established in children. However, numerous large-scale studies have shown infected children do not systematically exhibit lower SARS-CoV-2 viral load in the upper airway compared with adults[16,23–26], and ISG expression following infection does not appear significantly stronger in children when controlling for viral load. Thus, it remains unclear to what extent a pre-activated anti-viral state ultimately contributes to disparate clinical outcomes between children and adults who become infected.

This aside, our results suggest important elements of the adaptive immune response may be engaged more robustly in the upper airway of children. Specifically, we observed elevated gene expression markers of B cell and T cell activation, as well as cytokine production typically associated with T cell activation (such as IFNγ), in the upper airway of children. Intriguingly, Loske et al. also observed increased prevalence and activation of T cells as well as IFNγ expression in the upper airway of children in their single-cell study[21]. Vono et al. recently reported that children exhibited an elevated gene expression signature of B cell activation in the circulation compared to adults in the early days following symptom onset[16], and Yoshida et al. observed a striking increase in naïve lymphocytes in the circulation of children with COVID-19[17], which they speculated could reflect increased migration of B cells and T cells to the site of infection. Both our data and that of Loske et al. provide compelling evidence in support of this hypothesis. An early adaptive immune response to a novel pathogen in the upper airway of children, perhaps due to a more naïve immunological state, may therefore represent a critical factor in preventing progression to severe disease in children.

Finally, we observed evidence consistent with increased clearance of ciliated cells in children with SARS-CoV-2 infection, and a proportional increase in basal cells, which could be differentiating to restore homeostasis to the airway epithelium. Recent studies have found that ciliated cells are a major target for SARS-CoV-2 at the onset of infection[41,42]. It is thus conceivable that more effective turnover of infected ciliated cells and epithelial regeneration in the upper airway of children may limit the ability of the virus to migrate into the lower airway, where it can cause more severe disease.

Our study has several limitations that should be kept in mind: 1) a larger sample size would have increased the generalizability of the findings; 2) precise information on the timing of sample collection with respect to symptom onset was unavailable, although we limited our analysis to samples with viral load characteristic of the timeframe from just before symptom onset and up to ~6 days later[31]; 3) we did not have access to sequential data to investigate immune response dynamics over time; 4) we did not directly assess cell types present in the mucosa; and lastly, 5) the majority of subjects with COVID-19 had mild disease at the time of sampling and did not require hospitalization. Results may have differed if specimens from a greater proportion of

severely ill individuals had been available. However, this likely resulted in a more relevant comparison since relatively few children develop severe disease and the upper airway is no longer the principal site of pathology in severe disease.

Our study provides added perspective on several leading hypotheses regarding the molecular underpinnings of clinical outcome disparities between children and adults with COVID-19. Nevertheless, further study is warranted to understand why children are protected against severe disease from SARS-CoV-2, or β-coronaviruses more generally[43], as compared to several other respiratory viral pathogens.

## Methods

**Study design and clinical cohort.** The previously published adult cohort consisted of patients with acute respiratory illnesses tested for COVID-19 by RT-PCR at the University of California San Francisco (UCSF), leveraging leftover RNA extracted from clinical NP swab specimens[30]. The UCSF Institutional Review Board granted a waiver of consent under protocol #17-24056. For the analyses presented here, we leveraged this published adult dataset while supplementing it with pediatric samples similarly obtained at UCSF under the same protocol. Additionally, pediatric samples were obtained from patients tested for COVID-19 by RT-PCR from NP swabs at Children's Hospital Colorado (CHCO). CHCO specimens and data were obtained under Colorado Multiple Institutional Review Board protocols #20-0865, #20-1617 and #20-0972, which also granted a waiver of consent. All samples were collected between March and September 2020, prior to the availability of COVID-19 vaccines. Demographic and clinical data for all patients were obtained from a combination of electronic and manual abstraction of medical records at the respective institutions. Comprehensive sample metadata, including known clinical diagnoses, is available in Supplementary Data 1.

Children up to 19 years of age and adults at least 40 years of age were eligible for inclusion in the present analysis. Some samples were ultimately excluded based on sequencing metrics, as described in the following sections.

**Sample processing.** Excess clinical swab specimens were stored in viral transport media at $-80\,^{\circ}\text{C}$ in the respective Clinical Microbiology Laboratories. Specimens were thawed and 200 uL aliquots of specimen were added to 200 uL of DNA/RNA Shield (Zymo Research, Irvine, CA) in sterile 1.5 mL microtubes with appropriate biohazard precautions.

**Metagenomic RNA sequencing.** Specimens underwent RNA extraction and metagenomic sequencing, as previously described[30]. Briefly, RNA was extracted from 200 μL of specimen in DNA/RNA shield using bead-based lysis and the Zymo Pathogen Magbead kit (Zymo). We also processed negative control samples (water and HeLa cell RNA) to account for background contamination. All samples were spiked with RNA standards from the External RNA Controls Consortium (ERCC)[44]. Samples were DNase treated, depleted of cytosolic and mitochondrial rRNA using FastSelect (Qiagen, Germantown, MD), and reverse transcribed to generate cDNA. Sequencing libraries were constructed using the NEBNext Ultra II Library Prep Kit (New England Biolabs, Ipswich, MA). Libraries underwent 146 nucleotide paired-end sequencing on an Illumina Novaseq 6000 instrument.

**Metagenomic analysis of respiratory viruses.** Samples were processed through the CZ-ID pipeline (formerly called IDSeq)[45,46], which performs reference based alignment at both the nucleotide and amino acid level against sequences in the National Center for Biotechnology Information (NCBI) nucleotide (NT) and non-redundant (NR) databases, respectively, followed by assembly of the reads matching each taxon. We further processed the results for viruses with established pathogenicity in the respiratory tract[47]. We evaluated whether one of these viruses was present in a patient sample if it met the following three initial criteria: (i) at least 10 counts mapped to NT sequences, (ii) at least 1 count mapped to NR sequences, (iii) average assembly nucleotide alignment length of at least 70 bp.

Negative control (water and HeLa cell RNA) samples enabled estimation of the number of background reads expected for each virus, which were normalized by input mass as determined by the ratio of sample reads to spike-in ERCC RNA standards. Viruses meeting the initial criteria outlined above were then additionally tested for whether the number of sequencing reads aligned to them in the NT database was significantly greater than background. This was done by modeling the number of background reads as a negative binomial distribution, with mean and dispersion fitted on the negative controls. We estimated the mean parameter of the negative binomial for each taxon (virus) by averaging the read counts across all negative controls after normalizing by ERCC counts. We estimated a single dispersion parameter across all taxa using the functions glm.nb() and theta.md() from the R package MASS. We considered a sample to have a pathogenic respiratory virus detected by sequencing if the virus achieved an adjusted $p$-value < 0.05 after Holm's correction for all tests performed in the same sample.

We used the CZ-ID-calculated viral reads-per-million (rpM), based on the NT alignment, as a uniform measure of SARS-CoV-2 abundance across all samples. A value of 0.1 rpM was added to all samples with rpM < 0.1.

**Assignment of samples to comparator groups**. We divided the samples in each age cohort into three viral status groups: 1) samples with a positive clinical PCR test for SARS-CoV-2 were assigned to the "SARS-CoV-2" group; 2) samples with a negative PCR test for SARS-CoV-2 and no evidence of another pathogenic respiratory virus in the metagenomic sequencing were assigned to the "No Virus" group; and 3) samples with a negative PCR test for SARS-CoV-2 but another respiratory virus detected by sequencing were assigned to the "Other Virus" group.

We retained for analysis only samples in the SARS-CoV-2 groups with at least 10 rpM, roughly corresponding to PCR $C_t$ values below 30[30], to focus on cases with likely active viral replication, where a clear transcriptional response to the virus is expected to be found. This approach was based on the well-established correlation between viral load and recovery of actively replicating virus from respiratory specimens, specifically the finding that specimens with a PCR $C_t$ < 30 are associated with the ability to culture SARS-CoV-2[32–34]. Viral load above this threshold is typically characteristic of acute infection, within ~6 days of symptom onset[31].

**Human gene expression quantification**. Following demultiplexing, sequencing reads were pseudo-aligned with kallisto[48] (v. 0.46.1; including bias correction) to an index consisting of all transcripts associated with human protein coding genes (ENSEMBL v.99), cytosolic and mitochondrial ribosomal RNA sequences, and the sequences of ERCC RNA standards. Samples were retained for analysis if they had at least 400,000 estimated counts associated with transcripts of protein coding genes. Gene-level counts were generated from the transcript-level abundance estimates using the R package tximport[49], with the scaledTPM method.

**Differential expression (DE) analyses**. Genes were retained for each DE analysis if they had at least 10 counts in at least 20% of the samples included in the analysis. All analyses were performed with the R package limma[50], using quantile normalization and the voom method. The design formula for the comparisons within each age cohort was ~viral status, where viral status was either "SARS-CoV-2" or "No Virus". The design formula for the direct comparison between children and adults with SARS-CoV-2 infection was ~log₁₀(rpM) + age cohort, where age cohort was either "children" or "adults". DE $p$-values were adjusted using the Benjamini-Hochberg method within each comparison. Full DE results are available in Supplementary Data 2 and Supplementary Data 4.

**Gene set enrichment analyses (GSEA)**. Gene set enrichment analysis was based on Gene Ontology (GO) biological process pathway annotations[36], using the non-redundant version available through WebGestalt[51]. Only pathways with a minimum size of 10 genes and a maximum size of 1500 genes were retained for analysis. The analysis was performed using the fgseaMultilevel function in the R package fgsea[52], which calculates $p$-values based on an adaptive, multilevel splitting Monte Carlo scheme. The input consisted of all genes in the respective DE analysis, except for histone genes, pre-ranked by fold-change. The gene sets in Fig. 2 were manually selected to reduce redundancy and highlight diverse immune-related pathways and other relevant biological functions from among those with a Benjamini-Hochberg adjusted $p$-value < 0.05 in at least one of the three comparisons. Full results are provided in Supplementary Data 3 and Supplementary Data 5.

**In silico estimation of cell type proportions**. Cell-type proportions were estimated using the CIBERSORT X algorithm[39] based on single cell signatures derived from the human lung cell atlas[40]. Differences in estimated proportions between comparator groups were evaluated for statistical significance using a Mann-Whitney test with Holm's correction for multiple testing. Full results are provided in Supplementary Data 6.

**Regression of gene counts against viral load**. We performed robust regression of the limma-generated quantile normalized gene counts (log₂ scale) against log₁₀(rpM) of SARS-CoV-2 for $n = 100$ ISGs based on the "Hallmark interferon-alpha response" gene set in MSigDB (https://www.gsea-msigdb.org/gsea/msigdb),

as well as for selected B cell activation marker genes. The analysis was performed within each age cohort separately using the R package robustbase[53], which implements MM-type estimators for linear regression[54,55], the KS2014 setting, and the model: quantile normalized counts (log₂ scale) ~ log₁₀(rpM). Model predictions were generated using the R package ggeffects and used for display in the individual gene plots. Error bands represent normal distribution 95% confidence intervals around each prediction. Reported p-values for significance of the difference of the regression coefficient from 0 are based on a $t$-statistic. Reported $R^2$ values represent the adjusted robust coefficient of determination[56].

**Reporting summary**. Further information on research design is available in the Nature Research Reporting Summary linked to this article.

## Data availability

The raw sequencing data are protected due to patient privacy restrictions in the IRB protocols governing enrollment in this study under a waiver of consent. Researchers who wish to obtain the FASTQ files can contact the corresponding author in order to be added to the IRB protocols and sign a materials transfer agreement with UCSF ensuring secure storage of the data and its exclusive use for de-identified transcriptomic analyses. Processed gene counts have been deposited under NCBI GEO accession GSE179277. The published human lung single-cell datasets used for cell-type proportions analysis can be obtained through Synapse under accessions syn21560510 and syn21560511. The "Hallmark Interferon Alpha Response" gene set is available from MsigDB under accession M5911. Source data are provided with this paper.

## Code availability

Code for the cell-type proportions analysis and the robust regression analysis is available in the repository of the previously published study on the adult cohort: https://github.com/czbiohub/covid19-transcriptomics-pathogenesis-diagnostics-results.

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

## Acknowledgements
We would like to thank Sam Dominguez, MD, Ph.D. Kirk Harris, Ph.D. Aline Maddux, MD, Christina Osborne, MD, and Matthew Leroue, MD, for their input on study design and review of the data. We are grateful for funding support from the following sources: NHLBI K23HL138461-01A1 (C.R.L.); Chan Zuckerberg Biohub (C.R.L.); COVID-Child Health Research Award, Research Institute at Children's Hospital Colorado (P.M.M.); and philanthropic contributions from Mark and Carrie Casey, Julia and Kevin Hartz, Carl Kawaja and Wendy Holcombe, Eric Keisman and Linda Nevin, Martin and Leesa Romo, Diana Wagner, Jerry Yang and Akiko Yamazaki, and Three Sisters Foundation (C.R.L.).

## Author contributions
C.R.L., P.M.M., E.M., and A.T. designed the overall study with input from L.Am., B.D.W., J.L.D., and E.A.F.S. S.C., P.H.S., A.M.D., N.N., A.K., L.Am., and C.R.L. performed or oversaw sample acquisition, processing, or sequencing. N.S., L.M.L., H.R., K.R., K.M.W., V.S., B.D.W., L.Am., C.S., L.Ab., and C.R.L. contributed to sample metadata collation. E.M. and A.T. performed all analyses and data visualization. A.O.P. assisted with cell-type proportions analysis. E.M., A.T., and C.R.L. wrote the manuscript with input from co-authors.

## Competing interests
The authors declare no competing interests.
