## [Peer Review File · Nature Communications]

Upper airway gene expression reveals a more robust adaptive immune response to SARS-CoV-2 in childrenREVIEWER COMMENTS

Reviewer #1 (Remarks to the Author):

In this short article Mick and colleagues compare RNA-sequencing data from nasal swabs collected from children and adults with and without SARS-CoV2 infection. They demonstrate differences in gene expression between individuals with SARS-CoV2 infection vs no virus and differences between adults and children with SARS-CoV2 infection.

Overall the analyses are clear and generally appropriate for the data, and the messages are straightforward. The limited scope of the work seems appropriate for a short report as opposed to a full manuscript. The results are only descriptive and thus cannot be used to draw definitive conclusions about mechanisms of SARS-CoV2 infection in relation to clinical outcomes. Nevertheless, given the importance of publishing high quality data related to SARS-CoV2, it is understandable that this should be considered for a high impact journal.

My main concern has to do with potential differences in the cohorts that could be confounding the results, in particular disease severity. I'm not sure how the p-values are calculated in the supplementary demographics tables (the authors need to add this to the methods), but there appear to be some important differences in the adult SARS-CoV2 and peds SARS-CoV2 groups. There are 13% hospitalized adults and only 3% (1 person) peds. There are also 27% unknown in the adults which presumably could be hospitalized. I would strongly suggest that the author should consider a sensitivity analysis restricted only to the outpatient samples. Similarly there are big differences in the SARS-CoV2 vs no virus groups in both cohorts with most of the no virus individuals hospitalized. Inpatient vs outpatient is thus an important confounder for the authors primary analysis Fig 2A and should be addressed/commented on. Furthermore, what are these individuals hospitalized for?

Timing of symptom onset/infection compared to sample collection could also be very important as a confounder and the authors should note if they have any information about this or else list it as a weakness.

Additional comments in order:

Lines 71-79 – I'm not sure it is fair to characterize these as contradictory results. Both studies show an increase in some innate immune pathways in children, similar to your current study. In particular in the study by Koch et al (ref 21) when they restrict to children and adults with high viral load they saw genes upregulated "enriched for innate immune processes, including cellular response to IL-1 and inflammatory response". It might actually be worthwhile comparing the gene overlap observed in your study and these 2 studies to look for commonalities since many of the downstream signaling molecules responsible for the term enrichment for different innate immune pathways will be shared.

Figure 2 and lines 139-146 – Since the analysis hinges on these 3 pairwise DE/GSEA comparisons (peds SARSCoV2 vs no virus, adult SARSCoV2 vs no virus, peds SARSCoV2 vs adult SARSCoV2), the authors should do more to show the overlap and differences among these (eg venn diagrams of number of DEGs/number of pathways). It is notable that there are quite a few more genes and pathways in the 3rd comparison. The authors should also be more clear about how they subset to the pathways highlighted in figure 2, in particular since they are showing quite a few non-significant results among the 239 significant pathways that differ in peds vs adults. Are the results shown a fair representation of the findings?

Continuing on the above points and the comment about neutrophil activation in lines 149-150 – demonstrating the relative expression of some of these GO pathways in the 4 groups as boxplots/violin plots similar to figure 3 could be very illustrative, esp for some of the key pathways the authors wish to highlight.

Lines 182-195 and Fig 4 – How did the authors select which ISGs to display in Figure 4. Presumably there are also some ISGs that show a "lagging" response in adults instead. Is there a

preponderance of this finding among ISGs in children vs adults among the hundreds of ISGs? I don't think the authors can definitively conclude that the "lagging" response is responsible for the trend toward lower pathway expression observed in the GSEA results, at least they have not proven it here. Moreover, if this appears to be an important difference in peds vs adults, perhaps the authors should more comprehensively assess which aspects of the IFN response differ in kids vs adults.

Reviewer #2 (Remarks to the Author):

The study of Mick et al. aimed to investigate whether differences in the immune response of the upper airways between adults and children contributes to the severe disease cause of SARS-CoV2 infection that disproportionately affects adult patients. For this purpose, the authors collected clinical nasopharyngeal swab specimens from children, tested for an infection with SARS-CoV2 or another respiratory virus infection, determined the viral load, and analyzed gene by metagenomics RNA seq. The outcome of this analysis for the three groups (SARS-CoV2 positive, SARS-CoV2 negative, and positive for other resp. viruses) was compared to the gene expression analysis of a comparable and study carried out in adults previously published by the same group. In summary, the authors found that SARS-CoV2 infected children displayed upregulation of proinflammatory cytokines (e.g. IL-2, -4, IFNg, etc) as well as pathways related to B and T cell activation as compared to SARS-CoV2 infected adults, while expression of interferon-stimulated genes revealed no differences between the two cohorts. After performing an "in silico estimation of cell type proportions" the authors further concluded higher proportions of macrophages and lower proportions of ciliated cells in swap specimens of SARS-CoV2 infected children. Although the central question of the study is of particular interest, the study and its outcome is limited by several issues, which are in parts already mentioned by the authors in their own discussion.

- 1) The study of Mick et al. suffers from lack of novelty since Loske et al. (Nat Biotechnol 2021; doi: 10.1038/s41587-021-01037-9) carried out a comparable study with a technically advanced approach and reported an increased activity of the innate immune response of the upper airways of children.
- 2) Though the Loske study was performed with considerably less participants, it performed single cell RNA seq to assess gene expression of the cells found in the upper airways. This allows a much more precise analysis and interpretation of the gained data as compared to an "in silico estimation of cell type proportions" used in the present study, which could also explain the different outcomes of the respective studies. It is remarkable that these differences were not discussed in the present manuscript and that the Loske-study has not even been mentioned.
- 3) While it appears to be very difficult to determine the onset of infection, the fact that the present study did not include at least information on the onset of symptoms/disease (like the Loske study) makes interpretation of the data quite difficult - especially since gene expression provides only insight into the very actual biological processes, while the time course of infection continues for several days.
- 4) Since it was the aim of the study to contribute to understanding why adults disproportionately suffer from severe CoViD-19, it definitely makes sense to compare upper airway gene expression of children and adults with SARS-CoV2 expression and to perform this analysis at preferably early time point. However, the percentage of patients (adults as well as children) with a severe disease course is relatively low in this study (e.g. 4% ICU for adults and 0% ICU for children) and does not allow identification of gene expression differences between patients with mild or severe disease courses and to subsequent comparison/validation of such differences between the adult and children cohorts. Therefore, the significance of the present study regarding its central question is considerably limited.

Reviewer #3 (Remarks to the Author):

In this study authors analyzed the mucosal gene expression profile of children and adolescents with mild COVID-19 and leveraged an already published data set of adults with mild COVID-19 for comparison purposes. Authors also included a positive and negative viral group for each cohort comparison. They found that expression of ISGs was activated in both children and adults with COVID-19, however B-cells, T-cells, Th1, Th2 and other proinflammatory cytokine pathways (IFN- γ , TNF, IL-4 and IL-2) were greatly overexpressed in children compared to adults. In addition, in silico estimations of cell type proportions identified that ciliated cells were decreased in children while basal cells were increased. The study is well written and brings novel and interesting data to the scientific community. However, there are a number of aspects that need further clarification to allow better interpretation of the data.

Major comments

- When performing these types of analyses results are derived in relation to a “control” group. In this study the control group for both children and adults are the “no virus” group, which had significantly higher rates of hospitalization and ICU care as compared to COVID-19 patients (mostly diagnosed in the outpatient setting). In addition, there were no children included in the virus negative group that required ICU care, while this proportion increased to 11% in the adult cohort. How did authors adjust for these differences in severity between cases (COVID-19 patients) and controls (virus negative cohorts)?
- Do authors have any clinical information for the cohorts included in the study? i.e. underlying clinical conditions, duration of symptoms, use of steroids or immunomodulatory medications. In addition, in the viral negative group 36% of adults were hospitalized and 11% required ICU, while 82% of children in the viral negative group were hospitalized. What was the reason for hospitalization in these patients? Did they have pneumonia? It appears that information for 27% of adults and 3% of children is unknown. This information should be taken into consideration when analyzing the data and it is critical to be able to interpret the data appropriately.
- Authors could combine Supplemental Tables 1B and 1C and have it as a main table (The information of Table 1A is repeated in Tables 1B and C).
- Do authors have any information regarding viral loads for all other respiratory viruses? It is not infrequent that rhinoviruses are encountered incidentally in patients with other conditions, and viral loads may help to ascertain its true role in causing the disease.
- The supplementary tables are hard to reconcile with the manuscript as the labeling is incomplete and as currently provided not helpful to the reader.
- For Fig 2A, it appears that for some of the pathways there is tremendous overlap (i.e. regulation of innate immune response, response to interferon gamma, response to type-I interferon) between children and adults with COVID-19, yet authors concluded that some of these pathways (i.e. IFN-gamma) was greatly expressed in children than in adults. Please clarify.
- Authors mentioned that both in children and adults with COVID-19 the olfactory receptor gene expression was underexpressed that matched with loss in the sense of smell in these patients, but no clinical data is provided to support such conclusions.
- Figure 3 lacks the labeling indicating which color represents the pediatric cohort and which one the adult group.
- Authors did not perform multiparameter flowcytometry, so it is hard to conclude that the subtle differences in T-cells are due to expression rather than cell numbers (lines 166-167).
- While differences in monocyte/macrophage and neutrophils in adults with COVID-19 compare to adults with other viruses is evident, the overlap in the pediatric cohort is remarkable. I suggest authors modify the sentence in lines 170-171.
- The p-value for differences in dendritic cells was 0.17. As with monocytes and neutrophils the overlap is substantial. I suggest authors rephrase their conclusions (lines 172-173).
- The decreased of ciliated cells and parallel increased of basal cell is intriguing. One could also speculate that the proportional increased of basal cells is protective from severe COVID-19?
- IFI27 does not appear to correlate with viral loads in adults, and the R² for IFI6 is weak, while those correlations are clear in children. Overall it does not appear that children have a lagging ISG response in relation to SARS-CoV-2 loads but rather proportional.
- B-cell makers did not correlate with viral loads in children (rather than weakly correlated). Please modify. As authors mentioned the heterogeneity between pediatric patients was sizeable. Did authors collect duration of illness at enrollment?

- Additional limitations include the lack of clinical data (as mentioned above), cell types present in the mucosa at the time of infection or sequential data.

Minor comments

- I suggest that authors modify the first sentence in the abstract and introduction and instead of “rarely in children” they could state that the disease burden is lower. In the USA > 6 million children have been diagnosed with COVID-19 and > 600,000 have died because of the disease. Rates of infection have significantly increased these past months and on 10/21/21 children represented the 25% of all reported cases. Hospitalization rates have also increased and are ~ 2.2%.
- Under results (lines 111-112) authors should indicate if age is reported in years or months. For adults, intuitively the age reported would be in years but for children is not so clear.
- The information of Figures 1B and 1D could be included in a table format.
- Line 125: the adjusted p-value was calculated using Benjamini-Hochberg?
- Supplementary data 1 is not clearly labeled- It appears that a table labeled as 324975_0_data_set_5796421_qxdh2t has per gene information, however the number of transcripts do not match: 849 for the peds cohort and 848 for the adult cohort. In this other supplementary table (324975_0_data_set_5796420_qxdh2t) the number of genes are 14,966 for the pediatric cohort and 15773 for the adult cohort.

Reviewer #1 (Remarks to the Author):

In this short article Mick and colleagues compare RNA-sequencing data from nasal swabs collected from children and adults with and without SARS-CoV2 infection. They demonstrate differences in gene expression between individuals with SARS-CoV2 infection vs no virus and differences between adults and children with SARS-CoV2 infection.

Overall the analyses are clear and generally appropriate for the data, and the messages are straightforward. The limited scope of the work seems appropriate for a short report as opposed to a full manuscript. The results are only descriptive and thus cannot be used to draw definitive conclusions about mechanisms of SARS-CoV2 infection in relation to clinical outcomes. Nevertheless, given the importance of publishing high quality data related to SARS-CoV2, it is understandable that this should be considered for a high impact journal.

My main concern has to do with potential differences in the cohorts that could be confounding the results, in particular disease severity. I'm not sure how the p-values are calculated in the supplementary demographics tables (the authors need to add this to the methods), but there appear to be some important differences in the adult SARS-CoV2 and peds SARS-CoV2 groups. There are 13% hospitalized adults and only 3% (1 person) peds. There are also 27% unknown in the adults which presumably could be hospitalized. I would strongly suggest that the author should consider a sensitivity analysis restricted only to the outpatient samples.

Similarly there are big differences in the SARS-CoV2 vs no virus groups in both cohorts with most of the no virus individuals hospitalized. Inpatient vs outpatient is thus an important confounder for the authors primary analysis Fig 2A and should be addressed/commented on.

We appreciate the reviewer's attention to this important matter. Indeed, the No Virus groups in the two age cohorts comprise a large proportion of hospitalized patients (with non-viral conditions) whereas most patients in the SARS-CoV-2 groups are outpatients. To make sure this is clearly conveyed to readers, we have amended the text to state as follows:

Page 5, lines 109-111...113-115: "Most of the patients in the SARS-CoV-2 group in both age cohorts were tested as outpatients, indicative of an early/mild stage of disease (**Table 1**)... Patients in the No Virus group in both age cohorts were more likely to be hospitalized, with a higher proportion in the pediatric cohort (**Table 1; Supplementary Data 1**)."

This issue formed part of our motivation for juxtaposing an indirect comparison, using patients with and without SARS-CoV-2 infection within each age cohort, and a direct comparison of COVID-19 patients between the age cohorts. Each approach has its strengths and potential biases. The indirect comparison could be affected by differences in the level of clinical care but offers the benefit of controlling for baseline age-related differences and for any batch effects. The direct comparison focuses of course on the condition of interest, namely COVID-19, and is better matched in terms of clinical severity, but does not offer the perspective of a non-viral baseline.

Neither approach was designed to stand entirely on its own, rather, we reasoned that biological signals consistently revealed by both would represent the most reliable and relevant effects. Reassuringly, both approaches yielded overall consistent results with regard to several key immune pathways, and so our principal conclusions from **Fig. 2** taken as a whole are unlikely to be confounded. We also explicitly draw attention to a few inconsistent effects between the approaches, like the neutrophil and mast cell pathways:

Page 7, lines 152-154: “the stark disparity in neutrophil activation observed in the comparison to the No Virus groups was only weakly supported in the direct comparison, likely reflecting differences among the No Virus patients themselves.”

Although most COVID-19 patients in both age cohorts were outpatients, we agree that a sensitivity analysis within the direct comparison limited to outpatients is a beneficial way to evaluate the extent of potential confounding by clinical severity of COVID-19. We have thus repeated the differential expression analysis and GSEA on the outpatient subset of the cohort, comprising n=30 children and n=24 adults. We show the results, in comparison to the unrestricted version, in **Supp. Fig. 3**. Reassuringly, the outpatient-only results were very similar to those using the full cohort. While slight differences in the significance of certain pathways were observed, our principal conclusions remained unaltered. We have now added the following statement to the text:

Page 7, lines 160-163: “Importantly, we observed similar DE and GSEA results in a secondary analysis restricted only to outpatient children (n=30) and adults (n=24) with SARS-CoV-2 infection, suggesting that differences in the proportion of hospitalized patients and outpatients in each cohort did not bias the direct comparison (**Supplementary Fig. 3**).”

We have also clarified in the **Table 1** legend that p-values were calculated using a chi square test for dichotomous variables and using one-way ANOVA for continuous variables (age).

Furthermore, what are these individuals hospitalized for?

We have now added details on the admission diagnoses and preexisting comorbidities for all hospitalized patients in the second tab of **Supplementary Data 1**.

Timing of symptom onset/infection compared to sample collection could also be very important as a confounder and the authors should note if they have any information about this or else list it as a weakness.

Reliable information on the timing of symptom onset was unavailable to us. We note that reliable assessment and documentation of symptom onset specifically in young children can be challenging. We agree this is a weakness of our study and our limitations paragraph mentioned this.

Nevertheless, we took two approaches to attempt to control for stage of infection, which is the key relevant variable. First, we restricted the patients in the SARS-CoV-2 groups to those with viral load above a threshold equivalent to PCR $C_t < 30$. Studies have shown this range of viral load is typically observed in the nasopharynx in the ~7 day period ranging from ~1 day prior to symptom onset and up to ~6 days later (e.g., Kissler et al., N Engl J Med 2021, PMID 34941024). Second, we controlled for viral load in the direct differential expression analysis of children and adults with COVID-19, recognizing that viral load strongly correlates with stage of infection. We now explain this more clearly:

Page 5, lines 102-106: “we limited the samples in the SARS-CoV-2 group to those with at least 10 viral reads-per-million (rpm), comparable to PCR Ct values below 30. Viral load above this threshold is characteristic of acute infection, from just before symptom onset up to ~6 days later, and has been associated with recovery of actively replicating virus.”

Page 11, lines 261-264: “2) precise information on the timing of sample collection with respect to symptom onset was unavailable, although we limited our analysis to samples with viral load characteristic of the timeframe from just before symptom onset and up to ~6 days later.”

Additional comments in order:

Lines 71-79 - I’m not sure it is fair to characterize these as contradictory results. Both studies show an increase in some innate immune pathways in children, similar to your current study. In particular in the study by Koch et al (ref 21) when they restrict to children and adults with high viral load they saw genes upregulated “enriched for innate immune processes, including cellular response to IL-1 and inflammatory response”. It might actually be worthwhile comparing the gene overlap observed in your study and these 2 studies to look for commonalities since many of the downstream signaling molecules responsible for the term enrichment for different innate immune pathways will be shared.

We would like to clarify that our intention was to highlight contradictory results related mainly to the interferon response.

Pierce et al. (JCI Insight 2021, PMID 33822777) reported elevated ISG expression in children whereas Koch et al. (AJRCMB 2021, PMID 34731594) did not observe a difference. While our paper was in review/revision, two single-cell studies were published that also reported somewhat contradictory results as regards ISG expression. Loske et al. (Nat Biotech 2021, PMID 34408314) observed increased ISG expression in epithelial cells of children while Yoshida et al. (Nature 2021, PMID 34937051) observed the opposite, though the effect size in both cases was small. None of the studies directly controlled for viral load and most assessed a composite measure of ISG expression, which can be difficult to interpret. Moreover, the single-cell studies suggested that uninfected children already exhibit a pre-activated anti-viral state involving the interferon response. We therefore believe it is fair to say there is still a lack of consensus in the literature regarding age-related differences in ISG expression. To more precisely convey this point, we have amended the Introduction:

Page 3, line 61 - page 4, line 75: “Recent work has also begun to shed light on age-related differences in the immune response at the site of initial infection, the upper airway. It has been proposed that the upper airway of children is primed for viral sensing, exhibits a pre-activated anti-viral state, and/or engages a more robust innate immune response upon SARS-CoV-2 infection. However, numerous studies have found little to no evidence of a systematic difference between infected children and adults in the distribution of SARS-CoV-2 viral load in the nasopharynx, or in the kinetics of viral clearance, suggesting children do not control viral replication in the upper airway significantly better. Nevertheless, differences in the upper airway microenvironment and immune response could contribute to protection from severe disease in children in additional ways, for example, by limiting migration of the virus into the lower airway.

Several studies have placed particular focus on potential differences in interferon-stimulated gene (ISG) expression in the upper airway of children and adults, given its well-established importance as a front-line of anti-viral innate immunity. However, these studies reported some contradictory results, and none directly controlled for SARS-CoV-2 viral load on a gene-by-gene basis, highlighting the need for further investigation.”

We also now devote a significant portion of the Discussion to this important issue:

Page 9, line 218 - page 10, line 240: “Our analysis supports the conclusion that, when controlling for viral load, children and adults with SARS-CoV-2 infection both engage a pronounced interferon-stimulated gene (ISG) response in the upper airway that, as a whole, is of comparable magnitude. Our data further demonstrate that children exhibit a more gradual and proportional ‘dose response’ to viral load for a subset of prominent ISGs. These results are broadly in line with the findings of Koch et al., who also performed bulk RNA-sequencing on upper airway samples from children and adults and assessed a composite measure of ISG expression in patients with the highest viral load. Yoshida et al., who performed a single-cell RNA-sequencing study, also observed only subtle distinctions, with slightly stronger upregulation of composite ISG expression in epithelial cells of infected adults but slightly stronger upregulation in immune cells of infected children. In contrast, the single-cell analysis by Loske et al. showed somewhat elevated ISG expression in children, though neither single-cell study directly controlled for viral load.

Importantly, both single-cell studies provided evidence for a pre-activated anti-viral state in healthy children, characterized by elevated expression of upstream viral pattern recognition receptors and/or ISGs themselves. Our study was not well suited to examine this question since many patients in the No Virus groups were not healthy controls. It may indeed be the case that a pre-activated anti-viral state decreases the chance that a productive infection can be established in children. However, numerous large-scale studies have shown no systematic difference between infected children and adults in the distribution of SARS-CoV-2 viral load in the upper airway or in the kinetics of viral clearance, and ISG expression following infection does not appear stronger in children when controlling for viral load. Thus, it remains unclear to what extent a pre-activated anti-viral state ultimately contributes to disparate clinical outcomes between infected children and adults.”

We appreciate the suggestion to compare the DE gene overlap between our study and the Pierce et al. and Koch et al. studies. Unfortunately, full differential expression results were not provided in either of them, so we were unable to perform a systematic and unbiased comparison. Moreover, the pediatric sample size in the transcriptomic analyses in Pierce et al. was very small (n=6), no correction for viral load was attempted, and other than for ISG expression, many of the claims related to innate immunity were based on protein measurements.

As for Koch et al., in the preprint version, they indeed stated that children had higher expression of genes associated with innate immune processes alongside equivalent ISG expression. The recent journal publication retains the ISG result but now states that some innate immune pathways are upregulated in children while others are upregulated in adults, and no conclusion is made regarding IL-1 signaling. They shifted the focus of their conclusions to emphasize observations related to T cell signaling and function and neutrophil chemotaxis.

Figure 2 and lines 139-146 – Since the analysis hinges on these 3 pairwise DE/GSEA comparisons (peds SARSCoV2 vs no virus, adult SARSCoV2 vs no

virus, peds SARSCoV2 vs adult SARSCoV2), the authors should do more to show the overlap and differences among these (eg venn diagrams of number of DEGs/number of pathways). It is notable that there are quite a few more genes and pathways in the 3rd comparison.

We thank the reviewer for this thoughtful suggestion. We have now added as **Supp. Fig. 1** scatterplots relating the p-values for individual genes and pathways observed in the SARS-CoV-2 vs. No Virus comparisons within each age cohort. We further annotated the number of genes/pathways in each quadrant of the plot, conveying the information that would be contained in a Venn diagram, and labeled selected genes/pathways on the plots. In reference to this, we have added the following to the text:

Page 6, lines 122-126: “As expected, interferon-stimulated genes (ISGs) were prominent among the genes highly significant in both age cohorts (**Supplementary Fig. 1a**). Overall, children exhibited a considerably larger number of unique DE genes, including immune-related genes, despite a smaller sample size that would be expected to provide less statistical power (**Supplementary Fig. 1a**).”

One likely reason for the greater number of genes and pathways in the direct comparison is that this comparison additionally captures differences due to age alone, independent of SARS-CoV-2 infection. We now mention this in the text, as follows:

Page 6, lines 143-145: “We identified 5,352 differentially expressed genes at an adjusted p-value < 0.1 (**Supplementary Data 4**). Age differences unrelated to viral status likely contributed to the much larger number of DE genes in the direct comparison.”

The authors should also be more clear about how they subset to the pathways highlighted in figure 2, in particular since they are showing quite a few non-significant results among the 239 significant pathways that differ in peds vs adults. Are the results shown a fair representation of the findings?

We appreciate the opportunity to clarify. We selected the pathways in **Fig. 2** from among those that were significant in at least one of the 3 GSEA analyses, focusing on the major classes of immune-related pathways. Our aim was to juxtapose the findings of the SARS-CoV-2 vs No Virus comparisons within each age cohort and those of the direct comparison of the SARS-CoV-2 group between age cohorts, so we fixed the same pathways for display in both figure panels. Several pathways that were significant in one or more of the analyses in **Fig. 2a** were not significant in the analysis in **Fig. 2b**, which provides important context, though most of those pathways still generally trended in the same direction. As mentioned by the reviewer, there are many significant pathways in the direct comparison that we do not display. However, as we noted above, many of these pathways are unrelated to immune functions and presumably reflect fundamental age-related differences, which appear to us to be of lesser interest in the present context. The full results are provided as a supplementary file. We state:

Page 7, lines 158-159: “As expected, many developmental processes unrelated to infection also differed in the direct comparison between children and adults (**Supplementary Data 5**).”

And in the Methods section:

Page 16, lines 371-374: “The gene sets in **Fig. 2** were manually selected to reduce redundancy and highlight diverse immune-related pathways and other relevant biological functions from among those with a Benjamini-Hochberg adjusted p-value < 0.05 in at least one of the three comparisons. Full results are provided in **Supplementary Data 3** and **Supplementary Data 5.**”

Continuing on the above points and the comment about neutrophil activation in lines 149-150 - demonstrating the relative expression of some of these GO pathways in the 4 groups as boxplots/violin plots similar to figure 3 could be very illustrative, esp for some of the key pathways the authors wish to highlight.

Thank you for this suggestion. We now provide in **Supp. Fig. 2** violin plots showing normalized expression of several leading-edge genes driving the GSEA results for key pathways highlighted in the paper. We chose to display individual gene expression to provide readers with concrete and interpretable examples of what is driving the GSEA results. Composite pathway expression scores, which aggregate and average a large number of genes directly at the counts level, are often difficult to interpret and can obfuscate more than they reveal. The GSEA methodology itself is best suited to assess relative pathway expression but we cannot plot a single NES on the same scale across the 4 comparator groups, as GSEA is always performed between 2 groups.

Lines 182-195 and Fig 4 - How did the authors select which ISGs to display in Figure 4. Presumably there are also some ISGs that show a “lagging” response in adults instead. Is there a preponderance of this finding among ISGs in children vs adults among the hundreds of ISGs? I don’t think the authors can definitively conclude that the “lagging” response is responsible for the trend toward lower pathway expression observed in the GSEA results, at least they have not proven it here. Moreover, if this appears to be an important difference in kids vs adults, perhaps the authors should more comprehensively assess which aspects of the IFN response differ in kids vs adults.

We selected the genes for display following a global analysis of the relationship between ISG expression and viral load in children and adults, along the lines of the reviewer’s suggestion. We agree it would be preferable and more transparent to actually show this analysis in addition to the intriguing representative examples.

Specifically, we performed robust regression of normalized expression against SARS-CoV-2 viral load for a set of n=100 ISGs annotated in MSigDB. In **Fig. 4a** we now show a scatter plot relating the viral load slopes observed for each gene in the two age cohorts, and in **Fig. 4b** we show a similar scatter plot of the robust adjusted coefficient of determination (R^2). We also provide the full numerical results of the analysis as **Supplementary Data 7.**

This analysis demonstrates that several ISGs that are strongly correlated with viral load in adults (like *CXCL11*, *CXCL10*, *OASL*, *IFIT2*) show a similar pattern in children (similar R^2), with perhaps a slightly greater slope. However, for another group of genes (including *IFI6*, *IFI27*, *IFI44*, *HERC6*) the correlation with viral load in children is markedly better, as reflected by both greater R^2 and greater slope. For these genes, even adults with low viral load already exhibit elevated expression whereas the response in children is more gradual and proportional as a function of viral load. The opposite is rarely observed, with only *TRIM21* showing exceptionally better correlation with viral load in adults.

The genes *IFI6* and *IFI27* are among those in the leading-edge of the ‘response to type I interferon pathway’, which is trending toward lower expression in children with COVID-19 as compared to adults (as we now highlight in **Supp. Fig. 2d**). Based on our findings above, it is likely that the differences in ISG expression between children and adults in the lower viral load range are driving this apparent pattern, which once again illustrates the importance of taking viral load into consideration in all such analyses.

We now summarize these findings as follows:

Page 8, line 191 - page 9, line 208: “Finally, we wished to examine the effect of SARS-CoV-2 viral load on gene expression in pathways of interest within each age cohort. Expression of interferon-stimulated genes (ISGs) frequently correlates with viral load, as we observed in our previous analysis in adults. We performed robust regression to relate the expression of $n=100$ ISGs to viral load in children or adults with SARS-CoV-2 infection, and compared the resulting slopes and coefficients of determination (**Fig. 4a,b; Supplementary Data 7**). The ISGs that most strongly correlated with viral load in adults, such as *CXCL11* and *OASL*, exhibited overall similar patterns in children with slightly greater slopes (**Fig. 4a-c**). However, a subset of ISGs was considerably better correlated with viral load in children, most strikingly exemplified by genes such as *IFI6* and *IFI27* (**Fig. 4d**). While even adults with low viral load displayed elevated expression of these genes, the response in children was more gradual but caught up to the adults at higher viral loads. ISGs that shifted from an almost stepwise response to the virus in adults to a more proportional one in children were among the leading-edge genes that contributed to the trend toward lower interferon-response pathway expression in children in the GSEA results (**Fig. 2b; Supplementary Fig. 2d; Supplementary Data 5**). These findings suggest relatively subtle differences in ISG-specific regulation and/or cellular origins between children and adults that defy simple generalization. Such regulatory differences might also be reflected in the differential expression of certain interferon-regulatory factors (e.g., *IRF8*; **Supplementary Fig. 2c**.”

Reviewer #2 (Remarks to the Author):

Dear Editors, the study of Mick et al. aimed to investigate whether differences in the immune response of the upper airways between adults and children contributes to the severe disease cause of SARS-CoV2 infection that disproportionately affects adult patients. For this purpose, the authors collected clinical nasopharyngeal swab specimens from children, tested for an infection with SARS-CoV2 or another respiratory virus infection, determined the viral load, and analyzed gene by metagenomics RNA seq. The outcome of this analysis for the three groups (SARS-CoV2 positive, SARS-CoV2 negative, and positive for other resp. viruses) was compared to the gene expression analysis of a comparable and study carried out in adults previously published by the same group. In summary, the authors found that SARS-CoV2 infected children displayed upregulation of proinflammatory cytokines (e.g. IL-2, -4, IFN γ , etc) as well as pathways related to B and T cell activation as compared to SARS-CoV2 infected adults, while expression of interferon-stimulated genes revealed no differences between the two cohorts. After performing an “in silico estimation of cell type proportions” the authors further concluded higher proportions of macrophages and lower proportions of ciliated cells in swap specimens of SARS-CoV2 infected

children. Although the central question of the study is of particular interest, the study and its outcome is limited by several issues, which are in parts already mentioned by the authors in their own discussion.

1) The study of Mick et al. suffers from lack of novelty since Loske et al. (Nat Biotechnol 2021; doi: 10.1038/s41587-021-01037-9) carried out a comparable study with a technically advanced approach and reported an increased activity of the innate immune response of the upper airways of children.

Please note that we submitted our manuscript on August 5, 2021, before the study by Loske et al. was published on August 18, 2021. Moreover, we believe multiple studies, utilizing diverse clinical cohorts and applying diverse experimental and analytical approaches, together allow for a reliable consensus view to emerge.

In our revision, we naturally reference and discuss Loske et al. as well as several other important studies that have come out while our paper was in review/revision. These include Yoshida et al., another upper airway single-cell sequencing study whose results, like ours, aligned with some of the findings in Loske et al. but not with all of them. In particular, both our findings and those of Loske et al. support increased T cell activation and IFN γ production in children. However, our results diverge from the conclusion in Loske et al. regarding the expression of ISGs following infection. At the same time, we do not believe our study was best suited to assessing the question of a pre-activated anti-viral state in uninfected children, which both single cell studies proposed. However, we believe it is important to place this in the context of the observation made by multiple large-scale studies that infected children and adults exhibit comparable viral load distributions and viral clearance kinetics in the upper airway. We now state in the Discussion:

Page 9, line 218 - page 11, line 254: “Our analysis supports the conclusion that, when controlling for viral load, children and adults with SARS-CoV-2 infection both engage a pronounced interferon-stimulated gene (ISG) response in the upper airway that, as a whole, is of comparable magnitude. Our data further demonstrate that children exhibit a more gradual and proportional ‘dose response’ to viral load for a subset of prominent ISGs. These results are broadly in line with the findings of Koch et al., who also performed bulk RNA-sequencing on upper airway samples from children and adults and assessed a composite measure of ISG expression in patients with the highest viral load. Yoshida et al., who performed a single-cell RNA-sequencing study, also observed only subtle distinctions, with slightly stronger upregulation of composite ISG expression in epithelial cells of infected adults but slightly stronger upregulation in immune cells of infected children. In contrast, the single-cell analysis by Loske et al. showed somewhat elevated ISG expression in children, though neither single-cell study directly controlled for viral load.

Importantly, both single-cell studies provided evidence for a pre-activated anti-viral state in healthy children, characterized by elevated expression of upstream viral pattern recognition receptors and/or ISGs themselves. Our study was not well suited to examine this question since many patients in the No Virus groups were not healthy controls. It may indeed be the case that a pre-activated anti-viral state decreases the chance that a productive infection can be established in children. However, numerous large-scale studies have shown no systematic difference between infected children and adults in the distribution of SARS-CoV-2 viral load in the upper airway or in the kinetics of viral clearance, and ISG expression following infection does not appear stronger in children when controlling for viral load. Thus, it remains unclear to what extent a pre-activated anti-

viral state ultimately contributes to disparate clinical outcomes between infected children and adults.

This aside, our results suggest important elements of the adaptive immune response may be engaged more robustly in the upper airway of children. Specifically, we observed elevated gene expression markers of B cell and T cell activation, as well as cytokine production typically associated with T cell activation (such as IFN γ), in the upper airway of children. Intriguingly, Loske et al. also observed increased prevalence and activation of T cells as well as IFN γ expression in the upper airway of children in their single-cell study. Vono et al. recently reported that children exhibited an elevated gene expression signature of B cell activation in the circulation compared to adults in the early days following symptom onset, and Yoshida et al. observed a striking increase in naïve lymphocytes in the circulation of children with COVID-19, which they speculated could reflect increased migration of B cells and T cells to the site of infection. Both our data and that of Loske et al. provide compelling evidence in support of this hypothesis. An early adaptive immune response to a novel pathogen in the upper airway of children, perhaps due to a more naïve immunological state, may therefore represent a critical factor in preventing progression to severe illness in children.”

2) Though the Loske study was performed with considerably less participants, it performed single cell RNA seq to assess gene expression of the cells found in the upper airways. This allows a much more precise analysis and interpretation of the gained data as compared to an “in silico estimation of cell type proportions” used in the present study, which could also explain the different outcomes of the respective studies. It is remarkable that these differences were not discussed in the present manuscript and that the Loske-study has not even been mentioned.

As noted above, we did not reference Loske et al. because it had not been published at the time of our manuscript submission. We now discuss it, alongside additional relevant papers that were published in the intervening time. We also note that cell type deconvolution of bulk RNA-sequencing using single-cell markers is a widely practiced and useful approach, and while it is not without its limitations, so is single cell RNA-seq analysis (e.g, Squair et al., Nat Comms 2021, PMID 34584091). It was also one of several approaches we employed that generally converged on similar findings.

3) While it appears to be very difficult to determine the onset of infection, the fact that the present study did not include at least information on the onset of symptoms/disease (like the Loske study) makes interpretation of the data quite difficult - especially since gene expression provides only insight into the very actual biological processes, while the time course of infection continues for several days.

We agree that determining onset of infection is difficult, especially in children whose symptoms may be subtle and, in the case of very young children, cannot be readily communicated. This information was not rigorously tracked by treating providers at our study institutions, and we acknowledged that this is a weakness of our study in our limitations paragraph.

Nevertheless, we took two approaches to attempt to control for stage of infection, which is the key relevant variable. First, we restricted the patients in the SARS-CoV-2 groups to those with

viral load above a threshold equivalent to PCR $C_t < 30$. Studies have shown this range of viral load is typically observed in the nasopharynx in the ~7 day period ranging from ~1 day prior to symptom onset and up to ~6 days later (e.g., Kissler et al., N Engl J Med 2021, PMID 34941024). Second, we controlled for viral load in the direct differential expression analysis of children and adults with COVID-19, recognizing that viral load strongly correlates with stage of infection. We now explain this more clearly:

Page 5, lines 102-106: “we limited the samples in the SARS-CoV-2 group to those with at least 10 viral reads-per-million (rpm), comparable to PCR C_t values below 30. Viral load above this threshold is characteristic of acute infection, from just before symptom onset up to ~6 days later, and has been associated with recovery of actively replicating virus.”

Page 11, lines 261-264: “precise information on the timing of sample collection with respect to symptom onset was unavailable, although we limited our analysis to samples with viral load characteristic of the timeframe from just before symptom onset and up to ~6 days later.”

4) Since it was the aim of the study to contribute to understanding why adults disproportionately suffer from severe CoViD-19, it definitely makes sense to compare upper airway gene expression of children and adults with SARS-CoV2 expression and to perform this analysis at preferably early time point. However, the percentage of patients (adults as well as children) with a severe disease course is relatively low in this study (e.g. 4% ICU for adults and 0% ICU for children) and does not allow identification of gene expression differences between patients with mild or severe disease courses and to subsequent comparison/validation of such differences between the adult and children cohorts. Therefore, the significance of the present study regarding its central question is considerably limited.

Our aim was to identify early events in upper airway infection that may contribute to disparate clinical outcomes down the line. Severe COVID-19 in many respects is no longer a disease principally of the upper airway. Therefore, we believe the comparison of children and adults with mostly early/mild COVID-19 is the appropriate approach, and, incidentally, matches the approach taken by Loske et al.

Reviewer #3 (Remarks to the Author):

In this study authors analyzed the mucosal gene expression profile of children and adolescents with mild COVID-19 and leveraged an already published data set of adults with mild COVID-19 for comparison purposes. Authors also included a positive and negative viral group for each cohort comparison. They found that expression of ISGs was activated in both children and adults with COVID-19, however B-cells, T-cells, Th1, Th2 and other proinflammatory cytokine pathways (IFN-g, TNF, IL-4 and IL-2) were greatly overexpressed in children compared to adults. In addition, in silico estimations of cell type proportions identified that ciliated cells were decreased in children while basal cells were increased. The study is well written and brings novel and interesting data to the scientific community. However, there are a number of aspects that need further clarification to allow better interpretation of the data.

Major comments

- When performing these types of analyses results are derived in relation to a “control” group. In this study the control group for both children and adults are the “no virus” group, which had significantly higher rates of hospitalization and ICU care as compared to COVID-19 patients (mostly diagnosed in the outpatient setting). In addition, there were no children included in the virus negative group that required ICU care, while this proportion increased to 11% in the adult cohort. How did authors adjust for these differences in severity between cases (COVID-19 patients) and controls (virus negative cohorts)?

We thank the reviewer for drawing attention to this important matter. Indeed, the No Virus groups in the two age cohorts comprise a large proportion of hospitalized patients (with non-viral conditions) whereas most patients in the SARS-CoV-2 groups are outpatients. To make sure this is clearly conveyed to readers, we have amended the text to state as follows:

Page 5, lines 109-111...113-115: “Most of the patients in the SARS-CoV-2 group in both age cohorts were tested as outpatients, indicative of an early/mild stage of disease (**Table 1**)... Patients in the No Virus group in both age cohorts were more likely to be hospitalized, with a higher proportion in the pediatric cohort (**Table 1; Supplementary Data 1**).”

This issue formed part of our motivation for juxtaposing an indirect comparison, using patients with and without SARS-CoV-2 infection within each age cohort, and a direct comparison of COVID-19 patients between the age cohorts. Each approach has its strengths and potential biases. The indirect comparison could be affected by differences in the level of clinical care but offers the benefit of controlling for baseline age-related differences and for any batch effects. The direct comparison focuses of course on the condition of interest, namely COVID-19, and is better matched in terms of clinical severity but does not offer the perspective of a non-viral baseline.

Neither approach was designed to stand entirely on its own, rather, we reasoned that biological signals consistently revealed by both would represent the most reliable and relevant effects. Reassuringly, both approaches yielded overall consistent results with regard to several key immune pathways, and so our principal conclusions taken as a whole are unlikely to be confounded. None of our key findings rests solely on the comparison to the No Virus groups, and we also explicitly draw attention to a few inconsistent effects between the approaches (like the neutrophil and mast cell pathways).

In addition, we have now followed the advice of Reviewer 1 and performed a sensitivity analysis by restricting the direct comparison of children and adults with COVID-19 to outpatients only (n=30 children and n=24 adults). This is a beneficial way to evaluate the extent of potential confounding by clinical severity of COVID-19. We show the results, in comparison to the unrestricted version, in **Supp. Fig. 3**. Reassuringly, the differential expression and GSEA results using outpatients only were very similar to those using the full cohort. While slight differences in the significance of certain pathways were observed, our principal conclusions remained unaltered. We have now added the following statement to the text:

Page 7, lines 160-163: “Importantly, we observed similar DE and GSEA results in a secondary analysis restricted only to outpatient children (n=30) and adults (n=24) with SARS-CoV-2 infection, suggesting that differences in the proportion of hospitalized

patients and outpatients in each cohort did not bias the direct comparison (**Supplementary Fig. 3**).”

- Do authors have any clinical information for the cohorts included in the study? i.e. underlying clinical conditions, duration of symptoms, use of steroids or immunomodulatory medications. In addition, in the viral negative group 36% of adults were hospitalized and 11% required ICU, while 82% of children in the viral negative group were hospitalized. What was the reason for hospitalization in these patients? Did they have pneumonia? It appears that information for 27% of adults and 3% of children is unknown. This information should be taken into consideration when analyzing the data and it is critical to be able to interpret the data appropriately.

We appreciate the suggestion to include more clinical information in this study. We have now included the clinical diagnosis/reason for hospitalization for each patient, any underlying clinical conditions, and use of steroids/immunomodulatory agents in the second tab of **Supplementary Data 1**. We unfortunately do not have reliable data on the duration of symptoms, which we note in the limitations paragraph (see below).

We would like to emphasize again that the inclusion of the No Virus patients was mainly designed to provide a reference of upper airway gene expression in the absence of viral infection. We recognized the potential biases mentioned by the reviewer, and thus did not rely solely on the comparison to the No Virus groups for our key findings. Those were supported by the direct comparison between children and adults with COVID-19.

- Authors could combine Supplemental Tables 1B and 1C and have it as a main table (The information of Table 1A is repeated in Tables 1B and C).

As suggested, we have now combined Supplementary Tables 1A-C into main Table 1.

- Do authors have any information regarding viral loads for all other respiratory viruses? It is not infrequent that rhinoviruses are encountered incidentally in patients with other conditions, and viral loads may help to ascertain its true role in causing the disease.

Viral load (measured in reads-per-million, rpM) for the other virus samples was indeed also available through the CZ-ID (formerly IDSeq) pipeline. We now include sample-specific information regarding type of other virus detected and viral rpM in the metadata provided in the first tab of **Supplementary Data 1**. Of the 18 patients with rhinovirus across both age cohorts, only 1 had viral load below 10 rpM and most exhibited viral loads orders of magnitude higher than that.

- The supplementary tables are hard to reconcile with the manuscript as the labeling is incomplete and as currently provided not helpful to the reader.

We were unsure whether this comment refers to the supplementary tables we have now unified into a single main table (as suggested above) or to the supplementary data files. We were also unsure as to the specific labeling that seemed incomplete. In any case, we have now made a concerted effort to better organize and properly label all the supplementary materials.

- For Fig 2A, it appears that for some of the pathways there is tremendous overlap (i.e regulation of innate immune response, response to interferon gamma, response to type-I interferon) between children and adults with COVID-19, yet authors concluded that some of these pathways (i.e IFN-gamma) was greatly expressed in children than in adults. Please clarify.

We appreciate the opportunity to clarify. There are two pathways related to IFN γ . The first is called 'response to interferon-gamma', which includes downstream response genes that overlap to some extent with those in the 'response to type I interferon' pathway. Indeed, these pathways show robust activation in children and adults with COVID-19 compared to the respective non-viral groups (**Fig. 2a**) and only subtle differences in the direct comparison of children and adults with COVID-19 (**Fig. 2b**). However, there is a second pathway called 'interferon-gamma production', which focuses on upstream regulators and relevant cell type markers. This pathway is more strongly activated in children in both **Fig. 2a** and **Fig. 2b**. We have revised the text to more precisely delineate the significant pathways as follows:

Page 2, lines 34-36: "Children, however, demonstrated markedly greater upregulation of pathways related to B cell and T cell activation and proinflammatory cytokine signaling, including response to TNF and production of IFN γ , IL-2 and IL-4."

Page 6, lines 132-136: "Children, however, demonstrated stronger upregulation of the B cell activation, T cell activation, response to TNF, macrophage activation and phagocytosis pathways, as well as several cytokine production pathways typically associated with T cell activation, such as IL-2, IL-4, and IFN γ (**Supplementary Fig. 2a-c**)."

- Authors mentioned that both in children and adults with COVID-19 the olfactory receptor gene expression was underexpressed that matched with loss in the sense of smell in these patients, but no clinical data is provided to support such conclusions.

We would like to clarify that we did not base this statement on clinical data from our own patient cohort but rather on the general clinical observation reported in the literature that loss of sense of smell occurs both in adults and in children/adolescents with COVID-19, for which we provide relevant citations. To more clearly convey this point, we have amended the text to read:

Page 6, lines 138-140: "Both children and adults infected with SARS-CoV-2 exhibited downregulation of olfactory receptor gene expression ('sensory perception of chemical stimulus'), consistent with the loss of sense of smell that has been clinically observed across the age spectrum^{34,35}."

- Figure 3 lacks the labeling indicating which color represents the pediatric cohort and which one the adult group.

We thank the reviewer for pointing this out and have added the necessary labels to **Figure 3**.

- Authors did not perform multiparameter flowcytometry, so it is hard to conclude that the subtle differences in T-cells are due to expression rather than cell numbers (lines 166-167).

We agree that we cannot definitively draw this conclusion and have re-phrased as follows:

Page 8, lines 171-173: “In contrast, differences in estimated T cell proportions were much subtler (**Supplementary Fig. 4a**), suggesting the GSEA results may reflect distinctions in T cell identity and regulation and not only in cell number.”

- While differences in monocyte/macrophage and neutrophils in adults with COVID-19 compare to adults with other viruses is evident, the overlap in the pediatric cohort is remarkable. I suggest authors modify the sentence in lines 170-171.

We agree that it is not possible to definitively conclude whether the trend observed in adults recapitulates in children given the relatively small size of the pediatric Other Virus group and especially given that it is not a homogeneous group, but rather comprises various other viruses. Nevertheless, it is telling that approximately half of the pediatric Other Virus samples show higher proportions of macrophages, neutrophils and dendritic cells than the majority of SARS-CoV-2 samples. Coming on the backdrop of the results in the larger adult cohort, we believe we should comment on this. To better convey the appropriate caveats, we now state:

Page 8, lines 174-181: “We previously observed in our adult study that infection with SARS-CoV-2 was associated with blunted recruitment of macrophages and neutrophils to the upper airway as compared to other respiratory viruses. The pediatric Other Virus group was too small to definitively conclude whether this observation recapitulates in children, especially given the mix of different viruses represented, though it is notable that a substantial fraction of the samples in the other virus group exhibited markedly higher macrophage, neutrophil and dendritic cell proportions than most pediatric SARS-CoV-2 samples (**Fig. 3b-d**). Macrophage proportions did trend higher in children with SARS-CoV-2 infection compared with adults (**Fig. 3b**).”

- The p-value for differences in dendritic cells was 0.17. As with monocytes and neutrophils the overlap is substantial. I suggest authors rephrase their conclusions (lines 172-173).

We agree and have removed the statement regarding differences in dendritic cell proportions between children and adults with COVID-19, for which the adjusted p-value was 0.17.

- The decreased of ciliated cells and parallel increased of basal cell is intriguing. One could also speculate that the proportional increased of basal cells is protective from severe COVID-19?

We have now added this possibility:

Page 8, lines 187-190: “Recent studies have found that ciliated cells are a major target for SARS-CoV-2 at the onset of infection, and a possible interpretation of our results is that children are better able to clear infected ciliated cells in the nasopharynx, potentially helping them more effectively control the infection. It is also possible that the proportional increase of basal cells is protective.”

- IFI27 does not appear to correlate with viral loads in adults, and the R2 for IFI6 is weak, while those correlations are clear in children.

Overall it does not appear that children have a lagging ISG response in relation to SARS-CoV-2 loads but rather proportional.

We agree that referring to the response in children with lower viral loads as “lagging” compared to the adults with lower viral loads may give it a negative connotation, which is not necessarily the case and was not our intention. We have now replaced this term with the terms “proportional”, “gradual”, or “better correlated”, as suggested:

Page 9, lines 198-205: “However, a subset of ISGs was considerably better correlated with viral load in children, most strikingly exemplified by genes such as *IFI6* and *IFI27* (**Fig. 4d**). While even adults with low viral load displayed elevated expression of these genes, the response in children was more gradual but caught up to the adults at higher viral loads. ISGs that shifted from an almost stepwise response to the virus in adults to a more proportional one in children were among the leading-edge genes that contributed to the apparent trend toward lower interferon-response pathway expression in children in the GSEA results (**Fig. 2b; Supplementary Fig. 2d; Supplementary Data 5**).”

Page 9, line 220 - page 10, line 222: “Our data further demonstrate that children exhibit a more gradual and proportional ‘dose response’ to viral load for a subset of prominent ISGs.”

- B-cell makers did not correlate with viral loads in children (rather than weakly correlated). Please modify. As authors mentioned the heterogeneity between pediatric patients was sizeable. Did authors collect duration of illness at enrollment?

We have modified the text as suggested:

Page 9, lines 209-213: “In stark contrast to ISGs, the expression of B cell marker genes, such as *CD22* and *CD79A*, was entirely uncorrelated with viral load in children (**Fig. 4d**). These genes exhibited significant heterogeneity between patients, likely reflecting the timing of activation of the B cell response, but the fraction of children who were engaging the response at the time of sampling was substantially greater.”

We unfortunately did not have reliable information regarding the duration of illness available to us, which we note in our limitations paragraph:

Page 11, lines 261-264: “2) precise information on the timing of sample collection with respect to symptom onset was unavailable, although we limited our analysis to samples with viral load characteristic of the timeframe from just before symptom onset and up to ~6 days later.”

- Additional limitations include the lack of clinical data (as mentioned above), cell types present in the mucosa at the time of infection or sequential data.

We agree with the reviewer, and to address this we now provide additional clinical data including admission diagnoses and immunosuppressant medication administration, as noted above. We have also now specifically highlighted the other requested limitations in the Discussion, as follows:

Page 11, lines 264-265: “3) we did not have access to sequential data to investigate immune response dynamics over time; 4) we did not directly assess cell types present in the mucosa;”

Minor comments

- I suggest that authors modify the first sentence in the abstract and introduction and instead of “rarely in children” they could state that the disease burden is lower. In the USA > 6 million children have been diagnosed with COVID-19 and > 600,000 have died because of the disease. Rates of infection have significantly increased these past months and on 10/21/21 children represented the 25% of all reported cases. Hospitalization rates have also increased and are ~ 2.2%.

We appreciate this point and have modified the abstract and the Introduction as suggested:

Page 2, lines 27-28: “Unlike other respiratory viruses, SARS-CoV-2 disproportionately causes severe disease in older adults while disease burden in children is lower.”

Page 3, lines 44-46: “While infection with other respiratory viruses, such as influenza or respiratory syncytial virus, causes significant morbidity and mortality in both young children and older adults, severe COVID-19 occurs disproportionately in older adults.”

We would like to note, however, that as of 01/2022, there have been ~12,300 reported deaths due to COVID-19 in children and adolescents under 20-years-old worldwide, accounting for 0.4% of total deaths (<https://data.unicef.org/topic/child-survival/covid-19/>). This is clearly a tragic number in itself but much lower than 600,000.

- Under results (lines 111-112) authors should indicate if age is reported in years or months. For adults, intuitively the age reported would be in years but for children is not so clear.

We have revised this sentence to indicate that age is reported in years:

Page 5, lines 107-109: “The final dataset included 83 children (38 SARS-CoV-2, 34 No Virus, 11 Other Virus; median age 4 years, IQR 2-12) and 154 adults (45 SARS-CoV-2, 81 No Virus, 28 Other Virus; median age 62 years, IQR 47-71) (**Fig. 1a,b; Table 1; Supplementary Data 1**).”

- The information of Figures 1B and 1D could be included in a table format.

We appreciate the suggestion but prefer to convey this information graphically, as we believe it would be easier for readers to quickly get a sense of key cohort characteristics this way. The relevant information is available in **Supplementary Data 1** on a sample-by-sample basis.

- Line 125: the adjusted p-value was calculated using Benjamini-Hochberg?

Yes, adjusted p-values for differential expression were calculated using Benjamini-Hochberg. We have now added this to the Methods.

- Supplementary data 1 is not clearly labeled- It appears that a table labeled as 324975_0_data_set_5796421_qxdh2t has per gene information, however the number of transcripts do not match: 849 for the peds cohort and 848 for the adult cohort. In this other supplementary table (324975_0_data_set_5796420_qxdh2t) the number of genes are 14,966 for the pediatric cohort and 15773 for the adult cohort.

We apologize for the confusion and suspect that the supplementary data files were automatically renamed as part of the submission process, as these are not the names we gave them. We note that descriptions of all supplementary data files are provided on page 22. **Supplementary Data 1** now contains sample metadata. The gene-by-gene DE results are provided in **Supplementary Data 2** (for the SARS-CoV-2 vs No Virus comparisons) and **Supplementary Data 4** (for the direct comparison of children and adults with SARS-CoV-2 infection). This is now also referenced in the relevant sections of the Methods.

The number of genes in each of the three comparisons differs slightly because genes were retained for each DE analysis based on a minimal threshold of expression in a minimal fraction of the relevant samples. Since the samples included in each comparison were different, the genes included in the analysis also could differ slightly. The analysis within the pediatric cohort included 14,966 genes, the analysis within the adult cohort included 15,773 genes, and the pediatric vs adult analysis included 16,402 genes.

The 849 and 848 rows mentioned by the reviewer refer to pathways in the GSEA analysis, not individual genes. Here as well, the number of pathways represented in each analysis could differ slightly since the requirement for inclusion of a pathway was that at least 10 genes annotated in the pathway appear in the input DE results.

REVIEWERS' COMMENTS

Reviewer #1 (Remarks to the Author):

The authors have adequately address all of my prior comments and I feel the revised manuscript is improved and more clear and is ready for publication. In particular the secondary analysis restricted to outpatients is very reassuring to see. Further the additional of supplemental figures 1 and 2 help with the interpretation of the results. I have no further comments.

Reviewer #2 (Remarks to the Author):

I have thoroughly read the revised manuscript authored by Mick et al. and found the majority of points I have raised in my original review to be addressed. In detail, the authors have considered the Loske et al-Study, adapted the formulation of the study aim and took an attempt to handle the problem that the onset of infection/disease could not be exactly determined. Though this is of great importance to the evaluation and analysis of expression data (recorded at a given time point) against the background of the long lasting disease course, I appreciate the approach of the authors. From my point of view, the alterations of the authors markedly improved the original manuscript so that I can be considered for publication.

Reviewer #3 (Remarks to the Author):

Authors have addressed all comments and suggestions raised. I only have one remaining (minor) comment:

In the statement mentioning that "studies have found no differences between infected children and adults in SARS-CoV-2 viral loads" authors should mention that those studies did not include infants. In fact infants have been shown in different studies to have significantly higher viral loads than older children (PMID: 34850028; PMID: 32433729). Perhaps authors could say instead: "Excluding infants, numerous studies have found little to no evidence of difference between infect...."

Reviewer #1 (Remarks to the Author):

The authors have adequately addressed all of my prior comments and I feel the revised manuscript is improved and more clear and is ready for publication. In particular the secondary analysis restricted to outpatients is very reassuring to see. Further the additional of supplemental figures 1 and 2 help with the interpretation of the results. I have no further comments.

Thank you.

Reviewer #2 (Remarks to the Author):

I have thoroughly read the revised manuscript authored by Mick et al. and found the majority of points I have raised in my original review to be addressed. In detail, the authors have considered the Loske et al- Study, adapted the formulation of the study aim and took an attempt to handle the problem that the onset of infection/disease could not be exactly determined. Though this is of great importance to the evaluation and analysis of expression data (recorded at a given time point) against the background of the long lasting disease course, I appreciate the approach of the authors. From my point of view, the alterations of the authors markedly improved the original manuscript so that it can be considered for publication.

Thank you.

Reviewer #3 (Remarks to the Author):

Authors have addressed all comments and suggestions raised. I only have one remaining (minor) comment:

In the statement mentioning that "studies have found no differences between infected children and adults in SARS-CoV-2 viral loads" authors should mention that those studies did not include infants. In fact infants have been shown in different studies to have significantly higher viral loads than older children (PMID: 34850028; PMID: 32433729). Perhaps authors could say instead: "Excluding infants, numerous studies have found little to no evidence of difference between infect..."

We thank the reviewer for pointing out that infants have actually been shown to exhibit the highest viral load, further reinforcing our point that it is unlikely that children/infants are better able to achieve early control of viral replication in the upper airway, as proposed by the "pre-activated viral state" hypothesis. We have now added this to the Introduction:

Lines 64-69: "However, numerous studies have found little to no evidence of a systematic difference between infected children and adults in the distribution of SARS-CoV-2 viral load in the nasopharynx or in the kinetics of viral clearance^{16,23–26}, and a few studies have even shown infants exhibit the highest viral load^{27,28}. This suggests children are not significantly better able to achieve early control of viral replication in the upper airway."